# NRF2 and the Ambiguous Consequences of Its Activation during Initiation and the Subsequent Stages of Tumourigenesis

**DOI:** 10.3390/cancers12123609

**Published:** 2020-12-02

**Authors:** Holly Robertson, Albena T. Dinkova-Kostova, John D. Hayes

**Affiliations:** 1Jacqui Wood Cancer Centre, Division of Cellular Medicine, Ninewells Hospital and Medical School, University of Dundee, Dundee DD1 9SY, Scotland, UK; hr6@sanger.ac.uk (H.R.); A.DinkovaKostova@dundee.ac.uk (A.T.D.-K.); 2Wellcome Trust Sanger Institute, Wellcome Genome Campus, Cambridge CB10 1SA, UK

**Keywords:** NRF2, KEAP1, Cullin 3, ATF4, oxidative stress, reactive oxygen species, antioxidant, adaptation, glutathione, thioredoxin, NADPH generation, pentose phosphate pathway, proteasome, autophagy, drug metabolism, chemoprevention, chemotherapy, bioactivation, quinone-containing drugs, drug resistance, oncogene, tumour suppressor, initiation, progression, metastasis, recurrent disease, lung, oesophagus, liver, head and neck, stomach, bladder, colon, rectum

## Abstract

**Simple Summary:**

Transcription factor NRF2 controls expression of antioxidant and detoxification genes. Normally, the activity of NRF2 is tightly controlled in the cell, and is continuously adjusted to ensure that cells are protected against endogenous chemicals and environmental agents that perturb the intracellular antioxidant/pro-oxidant balance (i.e., redox) that must be maintained for them to grow and survive in an appropriate manner. This tight control of NRF2 is achieved by a repressor protein called KEAP1 that perpetually targets NRF2 protein for degradation under normal conditions, but is unable to do so when challenged with oxidants or thiol-reactive chemicals. In the context of cancer, it is well known that drugs that stimulate short-term and reversible activation of NRF2 can provide protection for a limited period against exposure to chemicals that cause cancer. However, it is also becoming widely recognised that permanent hyper-activation of NRF2 resulting from somatic mutations in the gene that encodes NRF2, or in genes associated with its degradation, is frequently observed in certain cancers and associated with poor outcome. In this article, we provide a critical overview of the literature describing the seemingly ambiguous contributions that NRF2 makes to the development of cancer. In particular, we describe the range of genetic and other mechanisms that are responsible for the upregulation of NRF2 in tumours, and highlight shortcomings in our knowledge of how frequently this occurs in different types of cancer. Moreover, we discuss how upregulation of NRF2 might aid the growth and survival of tumours, whether NRF2 upregulation in particular types of cancer is associated with mutations in specific oncogenes, and at what stage of cancer development this is likely to occur. Lastly, we discuss therapeutic strategies that have been proposed that selectively target tumours in which NRF2 is permanently activated with a view to overcoming NRF2-associated drug resistance.

**Abstract:**

NF-E2 p45-related factor 2 (NRF2, encoded in the human by *NFE2L2*) mediates short-term adaptation to thiol-reactive stressors. In normal cells, activation of NRF2 by a thiol-reactive stressor helps prevent, for a limited period of time, the initiation of cancer by chemical carcinogens through induction of genes encoding drug-metabolising enzymes. However, in many tumour types, NRF2 is permanently upregulated. In such cases, its overexpressed target genes support the promotion and progression of cancer by suppressing oxidative stress, because they constitutively increase the capacity to scavenge reactive oxygen species (ROS), and they support cell proliferation by increasing ribonucleotide synthesis, serine biosynthesis and autophagy. Herein, we describe cancer chemoprevention and the discovery of the essential role played by NRF2 in orchestrating protection against chemical carcinogenesis. We similarly describe the discoveries of somatic mutations in *NFE2L2* and the gene encoding the principal NRF2 repressor, *Kelch-like ECH-associated protein 1* (*KEAP1*) along with that encoding a component of the E3 ubiquitin-ligase complex *Cullin 3* (*CUL3*), which result in permanent activation of NRF2, and the recognition that such mutations occur frequently in many types of cancer. Notably, mutations in *NFE2L2*, *KEAP1* and *CUL3* that cause persistent upregulation of NRF2 often co-exist with mutations that activate KRAS and the PI3K-PKB/Akt pathway, suggesting NRF2 supports growth of tumours in which KRAS or PKB/Akt are hyperactive. Besides somatic mutations, NRF2 activation in human tumours can occur by other means, such as alternative splicing that results in a NRF2 protein which lacks the KEAP1-binding domain or overexpression of other KEAP1-binding partners that compete with NRF2. Lastly, as NRF2 upregulation is associated with resistance to cancer chemotherapy and radiotherapy, we describe strategies that might be employed to suppress growth and overcome drug resistance in tumours with overactive NRF2.

## 1. Introduction

The cap’n’collar (CNC) basic-region leucine zipper (bZIP) transcription factor NF-E2 p45-related factor 2 (NRF2, encoded by *NFE2L2*) is a master regulator of intracellular redox homeostasis because, in response to oxidative stress, it orchestrates induction of a battery of genes that serve to increase the antioxidant capacity of the cell. Since oxidative stress is associated with many common chronic debilitating ailments such as cancer, cardiovascular disease, diabetes mellitus, inflammatory disease, liver cirrhosis, lung fibrosis and neurodegenerative disease, it is not surprising that pharmacological agents that reversibly activate NRF2, and so alleviate oxidative stress, have been linked to the prevention or attenuation of many of these conditions [1]. Paradoxically, however, in certain types of cancer the irreversible genetic upregulation of NRF2 resulting from stochastic somatic activating mutations in *NFE2L2* or inactivating mutations in the gene encoding the repressor of NRF2, Kelch-like ECH-associated protein 1 (KEAP1), or that for its E3 ligase Cullin 3 (CUL3), is associated with progression of disease once it has been initiated [2]. In mammals, oxidative stress increases with age and this is accompanied by increased susceptibility to degenerative disease, which is associated with lower levels of intracellular antioxidants and the downregulation of NRF2 [3,4,5]. Considerable interest therefore exists around the interplay between degenerative disease, oxidative stress and NRF2, and how this can be exploited to improve health.

Although NRF2 was initially discovered by Yuet Wai Kan and colleagues as a human transcription factor that shared homology with the p45 subunit of NF-E2 and was capable of binding a tandemly-arrayed activator protein-1 (AP-1) recognition sequence (i.e., 5′-TGAGTCATGATGAGTCA-3′, with AP-1-binding sites underlined) in the β-globin gene locus, its involvement in directing cellular adaptation to oxidative stress was not immediately apparent [6,7]. This is possibly because many transcription factors, including AP-1, Forkhead box O (FOXO), peroxisome proliferator-activated receptor gamma coactivator 1-alpha (PGC-1α), nuclear factor-kappaB (NF-κB) and tumour protein p53 (TP53), control expression of antioxidant genes, and because NRF2 does not regulate directly the classic antioxidant enzyme superoxide dismutase (SOD) 1 or SOD2, unlike some of these other transcription factors [8]. Rather than being linked to the oxidative stress response, it was thought by various researchers that since the DNA sequence bound by NRF2 resembles an antioxidant responsive element (ARE, 5′-TGACNNNGC-3′) [9], also called an electrophile responsive element [10], the transcription factor might mediate induction of genes for drug-metabolising enzymes. Of particular significance, the DNA binding site also resembles a musculoaponeurotic fibrosarcoma (MAF) recognition element (MARE, 5′-TGCTGA^G^/_C_TCAGCA-3′) [11], and so the family of small MAF bZIP proteins, which can heterodimerize with NRF2, were considered as possible partners of NRF2 [12].

Based on a mouse gene knockout (ko) model, the first physiological role of Nrf2 was reported by Masi Yamamoto and colleagues to be that of mediating induction of drug-metabolising enzymes, in liver and small intestine, by the phenolic antioxidant butylated hydroxyanisole (BHA), the genes of which were known to contain ARE sequences in their regulatory regions; these included genes encoding NAD(P)H:quinone oxidoreductase 1 (NQO1, also called NAD(P)H dehydrogenase (quinone 1), and occasionally DT-diaphorase or menadione reductase) and various class Alpha, Mu and Pi glutathione S-transferase (GST) subunits [13]. Moreover, these workers proposed that NRF2 binds ARE/MARE sequences as a heterodimer with small MAF proteins, and this was later confirmed using compound small Maf knockout mice [14]. Careful examination of tissue extracts from Nrf2-ko mice revealed that the CNC-bZIP transcription factor controlled basal as well as inducible gene expression [13,15]. Later, the phenolic antioxidant ethoxyquin, the benzopyran coumarin and the related limettin (also called citropten), the butanolide α-angelica lactone, the dithiolethione oltipraz, the diterpenes cafestol and kahweol, the isothiocyanate sulforaphane (SFN), and indole-3-carbinol were all found to require the presence of Nrf2 to induce ARE-driven drug-metabolizing enzyme genes [16,17,18,19]. Since then, more potent NRF2-dependent inducers designed around a triterpenoid backbone, such as 1-[2-cyano-3,12-dioxooleana-1,9(11)-dien-28-oyl]imidazole and TBE-31, have been described [20,21]. See Figure 1 for the structures of these inducing agents.

Following recognition that Nrf2 regulates *Nqo1* and *Gst* subunit genes in the mouse, microarray analyses by Tom Kensler and others revealed that many other drug-metabolizing enzymes, including aldo-keto reductase (AKR), aldehyde dehydrogenase (ALDH), carbonyl reductase (CBR1), carboxylesterase (CES), epoxide hydrolase (EPHX), microsomal GST (MGST), prostaglandin reductase (PTGR) and UDP-glucuronosyltransferase (UGT), as well as ATP-binding cassette (ABC) drug-efflux pumps, are regulated by the CNC-bZIP transcription factor [22]. Around the same time, it emerged that NRF2 also controls genes encoding antioxidant proteins such as glutamate-cysteine ligase catalytic (GCLC) and modifier (GCLM) subunits, glutathione peroxidase 2 (GPX2), peroxiredoxin (PRDX1), cystine/glutamate antiporter xCT (SLC7A11), thioredoxin 1 (TXN1), thioredoxin reductase 1 (TXNRD1) and sulfiredoxin (SRXN1) [16,22,23,24,25]. In a similar vein, NRF2 also regulates genes for proteins involved in iron and heme metabolism, such as the ferritin heavy (FTH1) and light (FTL1) subunits, the iron transporter ferroportin (FPN1), biliverdin reductase B (BLVRB) and heme oxygenase 1 (HMOX1), and so likely limits the generation of reactive oxygen species (ROS) by Fenton chemistry [26,27]. In addition, NRF2-target genes include those encoding enzymes that generate NADPH, such as glucose-6-phosphate dehydrogenase (G6PD), isocitrate dehydrogenase 1 (IDH1), malic enzyme 1 (ME1) and 6-phosphogluconate dehydrogenase (PGD), and so supply reducing equivalents that allow oxidized glutathione and thioredoxin to be reduced and so maintain their antioxidant capacity [24,28,29,30]; of note, G6PD and PGD form part of the oxidative arm of the pentose phosphate pathway (PPP), whilst the non-oxidative arm includes transketolase (TKT) and transaldolase (TALDO1), both of which are also regulated by NRF2, though induction of these genes was shown by Hozumi Motohashi and colleagues to predicate on activation of growth factor signalling [31]. Because of its involvement in stress-inducible GSH and TXN1 synthesis and NADPH generation, these findings suggest NRF2 serves to maintain antioxidant capacity of cells. Consistent with this notion, it has been reported that in Nrf2-ko mouse embryonic fibroblasts (MEFs), the level of reduced glutathione (GSH) is only about 30% of that in wildtype MEFs, and that whilst treatment with the isothiocyanate SFN rapidly depletes intracellular GSH in both MEF lines, the ability of Nrf2-ko MEFs to regenerate GSH as an adaptive response to oxidative stress is severely impaired [32].

Besides its role in regulating detoxification and antioxidant genes, NRF2 controls expression of the scavenger receptors cluster of differentiation 36 (CD36) and macrophage receptor with collagenous structure (MARCO), which is consistent with it fulfilling a broadly protective role [33,34]. Additionally, Tom Kensler and colleagues discovered early on that NRF2 controls the expression of proteasome subunit (PSM) type A, B and C polypeptides associated with protein turn-over [22], and it is now also recognised to control many components of the autophagy system [35,36]. In addition, NRF2 controls the expression of various transcription factors, including the aryl hydrocarbon receptor (AHR), activating transcription factor (ATF) 3, ATF4, CCAAT/enhancer-binding protein beta (CEBPB), MafG, peroxisome proliferator-activated gamma (PPARγ, or PPARG), peroxisome proliferator-activated gamma coactivator 1β (PPARGC1B) and retinoid X receptor alpha (RXRα, or NR2B1) [37,38,39]. NRF2 also controls its own expression as well as that of its repressor KEAP1 [39].

Activation of NRF2 allows cross-talk with a number of signalling processes. These include Ca^2+^-signalling by regulation of the gene encoding transient receptor potential cation channel, subfamily A, member 1 (TRPA1), resulting in triggering of pro-survival pathways [40]. NRF2 also cross-talks in a bidirectional fashion with Notch signalling [41]. Moreover, NRF2 has been reported to influence hedgehog signalling in both a positive [42] and negative [43] manner. Table 1 provides a list of genes regulated by NRF2.

## 2. Historical Perspective on NRF2 and Cancer Chemoprevention

During early characterisation of the Nrf2-ko mouse, many small molecules that were found to require the CNC-bZIP transcription factor in order to induce *Nqo1* and *Gsta1* genes had already been classed as cancer chemopreventive ‘blocking’ agents because they can prevent or retard the initiation of carcinogenesis in rodents if administered prior to treatment with a chemical carcinogen [44]. In a seminal review written in 1966 on agents that can inhibit chemical carcinogenesis [45], the origins of chemoprevention were traced back to the pioneering work of Isaac Berenblum and colleagues who published a series of papers from 1929 onwards showing that low doses of dichloroethyl sulphide (mustard gas) could protect against skin tumourigenesis produced by topical application of tar to mice [46,47]. Since mustard gas is highly toxic and the demonstration of its effects limited to skin-painting experiments, these findings were of limited public health relevance as they could not be applied to humans. Subsequently, Herbert Crabtree reported that a range of relatively innocuous compounds could protect against skin carcinogenesis caused by 3,4-benzo[*a*]pyrene [48,49] and, most intriguingly, linked the ability of these agents to inhibit induction of tumours to alterations in sulfur metabolism [48]. The notion that chemoprevention extended beyond skin cancer, and could not, therefore, be simply attributed to physical shielding against absorption of the carcinogen across the skin, was revealed by several research groups, including that of Elizabeth and James Miller, who showed that administration of a low dose of various polycyclic aromatic hydrocarbons protects against liver carcinogenesis initiated by 3-methyl-4-dimethylaminoazobenzene or 2-acetylaminofluorene [50,51,52], results that indicated chemopreventive agents exerted systemic effects.

As chemical carcinogens typically have to be catalytically converted to an unstable reactive electrophilic intermediate in order to exert their mutagenic and carcinogenic effects [53], in the case of polycyclic aromatic hydrocarbons by forming an epoxide [54,55], the laboratory of Lee Wattenberg investigated whether chemopreventive agents might alter drug metabolism in target organs and so decrease the formation of electrophilic ultimate carcinogens. For example, these workers showed that treatment of rats and mice with various flavones could increase the hydroxylase activity of phase I cytochrome P450 (CYP) enzymes towards benzo[*a*]pyrene in liver and lung, and that pre-treatment of A/HeJ mice with β-naphthoflavone, and to a lesser extent quercetin, before p.o. administration of benzo[*a*]pyrene greatly diminished the number of lung tumours subsequently observed at autopsy [56,57]. Later, oltipraz was reported to protect against lung tumourigenesis initiated by various carcinogens including benzo[*a*]pyrene, diethylnitrosamine and uracil mustard [58].

The potential therapeutic value of chemoprevention received a major credibility boost with the discovery that synthetic antioxidants such as BHA, butylated hydroxytoluene (BHT) and ethoxyquin inhibited forestomach carcinogenesis in the mouse initiated by benzo[*a*]pyrene or 7,12-dimethylbenz[*a*]anthracene (DMBA) [59], and that BHT inhibited liver and mammary carcinogenesis in the rat initiated by 2-acetylaminofluorene [60], principally because BHA and BHT are encountered as food preservatives [61]. Moreover, benzyl isothiocyanate, indoles, coumarins and α-angelicalactone, which are also commonly found in the human diet, were also found to inhibit chemical carcinogenesis [62,63].

With the finding that synthetic antioxidants can protect against carcinogenesis in a range of tissues, came the realisation that they can induce enzymes that catalyse phase I drug-oxidation/reduction reactions as well as phase II drug-conjugation reactions. Indeed, it was discovered that chemopreventive agents augment the activities of EPXH1, NQO1 and AKR isoenzymes as well as GST and UGT isoenzymes in various tissues [64,65,66,67,68,69,70,71]. Importantly, the increase in GST activity stimulated by BHA was shown to result from an increase in abundance in mRNA for the GST subunits, rather than merely an increase in catalytic activity [72,73], and this is true too for the increases in EPXH1, NQO1, AKRs and UGTs. In the case of benzo[*a*]pyrene, it is activated by CYP1A1 and CYP1B1 to the ultimate carcinogen 7β,8α-dihydroxy-9α,10α-oxy-7,8,9,10-tetrahydrobenzo[*a*]pyrene, which in rodents that have been fed a diet containing BHA is principally inactivated by EPXH1 and UGT isoenzymes [74], with GST-catalysed conjugation of GSH with the benzo[*a*]pyrene diol epoxide playing a relatively minor role [75] (Figure 2A). It should be noted also that work in human lung cell lines suggests that AKR isoenzymes participate in the metabolic activation of benzo[*a*]pyrene [76].

Cancer chemopreventive agents provide protection against exposure to naturally occurring carcinogens, with a good example being provided by aflatoxin B_1_ (AFB_1_), produced by the fungus *Aspergillus flavus*, which is a potent hepatocarcinogen but displays selective species toxicity [77]; rats are relatively sensitive to AFB_1_ toxicity, whereas mice are resistant. Treating rats with ethoxyquin or oltipraz was found to protect against AFB_1_-initiated hepatocarcinogenesis [78,79]. In this case, protection against the mycotoxin is attributed to induction of genes encoding GSTA5 (originally called Yc_2_), a class Alpha GST subunit that in the dimeric enzyme conjugates the genotoxic AFB_1_
*exo*-8,9-epoxide with GSH, and AKR7A1 (originally called AFAR), which catalyses NADPH-dependent reduction of the cytotoxic AFB_1_-dialdehyde to an AFB_1_-dialcohol [80,81,82,83,84,85,86] (Figure 2B). Interestingly, mouse liver constitutively expresses high levels of GST A3-3 (previously called a YcYc homodimer), which has high activity towards AFB_1_
*exo*-8,9-epoxide and is thought to be responsible for the intrinsic resistance of the mouse to AFB_1_ hepatocarcinogenesis [87]. Indeed, knockout of *Gsta3* renders mice sensitive to the hepatotoxic effects of the mycotoxin [88].

## 3. Global Knockout of NRF2 in the Mouse Results in Diminished Intrinsic Resistance to Chemical Carcinogenesis and Impairs the Efficacy of Cancer Chemoprevention

As described above, small molecules that inhibit the initiation of carcinogenesis have been designated cancer chemopreventive ‘blocking’ agents, whereas those that inhibit the later post-initiation stages of cancer have been called ‘suppressing’ agents [44]. Following the discovery that NRF2 regulates both the basal and inducible expression of genes encoding drug-metabolising enzymes capable of detoxifying carcinogenic xenobiotics, various research groups investigated whether loss of the CNC-bZIP transcription factor might sensitise mice to tumourigenesis and diminish the effectiveness of chemopreventive agents in gastric, bladder, skin, gastrointestinal tract, oral, mammary, lung and liver cancer. Such studies revealed that Nrf2-ko mice are more sensitive to DNA damage caused by various genotoxic compounds, such as benzo[*a*]pyrene, than wildtype mice [89]. Moreover, knockout of NRF2 in the rat renders them more sensitive to formation of AFB_1_-N^7^-guanine DNA adducts in the liver when exposed to the mycotoxin [90].

The first chemoprevention study to focus on Nrf2-ko mice revealed that whilst 48 h pre-treatment of the mutant mice with oltipraz could decrease the number of benzo[*a*]pyrene-initiated forestomach neoplastic lesions by approx. 55%, oltipraz did not decrease tumour numbers in Nrf2-ko mice [17]. Furthermore, benzo[*a*]pyrene produced a higher tumour burden in the forestomach of Nrf2-ko mice than in wildtype mice. A similar loss of chemoprevention in Nrf2-ko mice was found against benzo[*a*]pyrene-initiated forestomach cancer using SFN [19].

In a subsequent study of bladder chemoprevention in Nrf2-ko mice, which used *N*-butyl-*N*-(4-hydroxybutyl)nitrosamine (BBN) in the drinking water to initiate carcinogenesis, it was found that feeding wildtype mice an oltipraz-containing diet a week before exposing them to BBN for 8 weeks, with oltipraz treatment for the duration of the experiment, reduced the incidence of urinary bladder cancer to approx. 50% in wildtype mice, but this protection was not afforded to Nrf2-ko mice treated in an identical manner [91]. Interestingly, the incidence of BBN-initiated bladder cancer was significantly higher in Nrf2-ko mice than age-matched wildtype mice. Moreover, the Nrf2-mediated protection against bladder cancer was attributed to increased metabolism of BBN by Ugt1a6.

Later, in a murine skin cancer model that employed a single dose of DMBA followed by repeated applications over a 25-week period of the tumour promoter 12-*O*-tetradecanoylphorbol-13-acetate (TPA), Nrf2-ko mice developed more tumours than their wildtype counterparts and continuous topical application of SFN that was started prior to DMBA treatment did not diminish tumour numbers, whilst in wildtype mice SFN treatment decreased the incidence of tumours and the numbers per mouse over the duration of the experiment [92]. Importantly, this observation suggested that activation of Nrf2 results in a blunting of both the initiation and promotion stages of tumourigenesis [92], and is further supported by studies employing ultraviolet radiation (UVR) as a complete carcinogen, using either pure SFN [93], or broccoli extracts as a source of SFN [94] or its biogenic precursor glucoraphanin [95]; it should, however, be noted that in the studies of the effects of SFN on UVR tumourigenesis the dependence of SFN on Nrf2 has not been investigated. 

In colon cancer, the possibility that pharmacological activation of NRF2 protects against tumourigenesis has been investigated using a colitis-associated carcinogenesis model that involved administration of a single dose of azoxymethane (AOM) followed by exposure for 10 weeks to dextran sulfate sodium (DSS). Utilizing this regimen, pre-treatment with the Nrf2 activator cinnamaldehyde one week before the administration of AOM, and thereafter maintained throughout DSS exposure, resulted in the number of colon tumours being reduced to approx. 50% in wildtype mice but no such reduction was observed in Nrf2-ko mice [96]. Interestingly, these workers did not report a higher incidence of tumours in Nrf2-ko mice when compared with wildtype mice.

In a model of oral carcinogenesis that involved provision of drinking water containing 4-nitroquinoline 1-oxide (4NQO) for 16 weeks followed by ordinary drinking water for 8 weeks, it was found that topical application of SFN to the tongue three times per week throughout the experiment decreased oral carcinogenesis to about 63% in wildtype mice [97]. By contrast, these workers found that SFN provided no protection against oral carcinogenesis in Nrf2-ko mice [97]. Interestingly, Nrf2-ko mice are more sensitive to 4NQO-initiated carcinogenesis than wildtype mice [98].

Chemoprevention of DMBA-induced mammary cancer in rats has been reported using various blocking agents that are known to activate Nrf2, including SFN [99], and although an NRF2-ko rat is available it is currently not known if such protection is abolished in the mutant animal. Nrf2-ko mice have been found to be more sensitive to DMBA-induced mammary cancer than their wildtype counterparts [100], but little has been reported about the ability of chemopreventive agents to protect against mammary tumourigenesis in Nrf2-ko animals.

Early experiments into chemoprevention of lung cancer indicated that β-naphthoflavone and oltipraz can protect against initiation of tumourigenesis by benzo[*a*]pyrene, diethylnitrosamine and uracil mustard [57,58] but it is not known if this can be attributed to activation of Nrf2. More recent experiments have shown that the flavone chrysin, when administered to mice either before or 4 weeks after treatment with benzo[*a*]pyrene diminished histopathology markers of lung cancer, but again it is not known if this protection was mediated by Nrf2 [101]. Furthermore, a curcumin-related compound called bis[2-hydroxybenzylidene]acetone (BHBA) has been shown to protect female A/J mice against the formation of lung adenocarcinoma when given for two weeks prior to the administration of vinyl carbamate, but even though BHBA was shown to induce ARE-driven luciferase activity, it remains uncertain whether in this instance chemoprevention is mediated by Nrf2 [102]. Using an ethyl carbamate-induced lung carcinogenesis model, Nrf2-ko mice have been shown to be substantially more sensitive than wildtype mice to the early stages of tumourigenesis but, surprisingly, seem to be more resistant to the later stages of lung tumourigenesis than their wildtype counterparts, suggesting that whilst Nrf2-target genes antagonise initiation of chemical carcinogenesis they facilitate the later transformation of benign adenoma to adenocarcinoma that may entail mutation of *Kras* [103]. Interestingly, the vinyl carbamate-initiated lung adenocarcinomas in Nrf2-ko mice were found to be larger than those in wildtype mice and were characterized by infiltration of tumour-promoting immune cells as evidenced by overexpression of genes for cytokines and antigen presentation, including Csf1, Cxcl1, Cxcl12 and Ccl9 [104]. Support for the notion that Nrf2 differentially influences the various stages of carcinogenesis has come from the fact genetic activation of Nrf2, using the Keap1-floxed hypomorph (i.e., *Keap1-*knockdown or *Keap1^FA/FA^*) mouse, protects mice against tumourigenesis, probably by upregulating genes encoding enzymes that detoxify the ultimate carcinogen vinyl carbamate epoxide, but that the tumours obtained from the *Keap1^FA/FA^* mice grew more robustly when transplanted into nude mice than those from *Keap1^+/+^* mice [105]. Consistent with this proposal, in *Kras^LSL-G12D/+^* mice, which harbour oncogenic Kras, pharmacological upregulation of Nrf2 by treatment with SFN after lung tumourigenesis had been initiated has been found to result in a modest 1.35-fold increase in the number of tumours observed on the surface of the lungs [106], though it should be noted that this is possibly controversial as prolonged treatment with SFN has also been reported not to increase cancer in Kras-based models of lung tumourigenesis [107].

The study of chemoprevention of liver cancer has been complicated by the fact that the classic hepatocarcinogen AFB_1_ displays marked species selectivity: the rat is sensitive to AFB_1_-hepatocarcinogenesis and can be protected by chemopreventive blocking agents, whereas the mouse is resistant. NRF2-null rats are highly sensitive to the genotoxic effects of AFB_1_ and have been found to die of liver cirrhosis rather than develop hepatomas (Dr Keiko Taguchi and Professor Masi Yamamoto, personal communication). Hepatocarcinogenesis can be studied in the mouse using CCl_4_, DMBA or diethylnitrosamine (DEN). Nrf2-ko mice are substantially more sensitive to CCl_4_ than wildtype mice, and pharmacological or genetic activation of the CNC-bZIP transcription factor can protect against toxicity [108,109]. Little has been reported about the effect of Nrf2 on the toxic effects of DMBA in mouse liver. However, rather surprisingly, it has been found that in the mouse, Nrf2 upregulation resulting from somatic mutation in the *Nfe2l2* gene is necessary for DEN-induced liver carcinogenesis, and thus Nrf2-ko mice are resistant to the carcinogenic effects of DEN [110].

Whilst not documented, it seems probable that tissues in which Nrf2-ko mice are more sensitive than wildtype mice to chemical carcinogenesis reflect the extent to which the CNC-bZIP transcription factor controls the basal expression of drug-metabolising enzymes that detoxify the chemical carcinogen employed to initiate cancer. These intrinsic variables may in turn be influenced by environmental factors such as diet. Similarly, the robustness of chemoprevention will likely depend on the degree to which Nrf2 can increase the activity of detoxification enzymes and the duration of this increase in activity. Besides these pharmacology-associated variations in the host, once initiated, the premalignant adenomas, adenocarcinomas and malignant cells will exert a wide range of host-tumour cell variations that will reflect the types of somatic mutations they have acquired during evolution of the tumour. Presumably, these variations reflect the tumour type and the environment in which they arise.

## 4. Cancer Chemopreventive Blocking Agents Possess Thiol Reactivity

Up until the discoveries of NRF2 and KEAP1, the mechanism by which cancer chemopreventive agents induce gene expression was the subject of considerable conjecture. For example, as AHR-mediated induction of *CYP1A1* by polycyclic aromatic hydrocarbons provided the earliest paradigm for the regulation of drug-metabolising genes by xenobiotics, some thought it was likely that ligand binding to a receptor would be necessary for chemopreventive agents to induce genes encoding GSTs and NQO1. However, Paul Talalay and colleagues pointed out that whilst the anti-cancer compounds that induce *Nqo1* in murine Hepa1c1c7 cells varied enormously in structure, they were all Michael reaction acceptors and thus electrophilic in nature, and proposed that this was the trigger for induction of *Nqo1*, rather than some receptor-defined stereochemical property [111,112]. Besides Michael acceptors, these workers later identified a diverse range of inducing agents with electrophilic properties including diphenols, phenylenediamines, quinones, dithiolethiones, isothiocyanates, H_2_O_2_, hydroperoxides, mercaptans, trivalent arsenicals and heavy metals that activated ARE-driven gene expression [113]. Because of their ability to react with nucleophilic sulfhydryl groups, many chemopreventive agents can form conjugates with GSH and would be capable of modifying thiol groups in proteins [114,115], and many of these agents were found to be substrates for GST [116,117]. As chemopreventive agents increase the abundance of enzymes that detoxify electrophiles, along with levels of GSH, the mode of adaptation they elicited was dubbed the ‘electrophile counterattack response’ [118].

## 5. Repression of NRF2 by KEAP1 Requires Two Binding Motifs in the Transcription Factor

Identification of KEAP1 as a protein that binds the *N*-terminal NRF2-ECH homology (Neh) 2 domain of NRF2, and inhibits basal and inducible ARE-driven gene transcription, represented a milestone in the chemoprevention field [119]: NRF2 contains seven domains, Neh1-Neh7 (Figure 3). Crucially, KEAP1 binds NRF2 through its Kelch-repeat domain [120], and as it is a dimeric protein it contains two NRF2-binding sites [121,122]. KEAP1 contains five domains, namely, an N-terminal region (NTR), a Broadcomplex, Tramtrack, Bric-à-brac (BTB) dimerization domain, an intervening region (IVR) a six-bladed Kelch-repeat domain, and a C-terminal region (CTR). Notably, two separate amino acid sequences within the Neh2 domain of NRF2 can bind to the KEAP1 dimer, called DLG and ETGE motifs (representing amino acids 29–31 and 79–82, respectively), with the ETGE motif possessing a much greater affinity for the Kelch-repeat domain of KEAP1 than the DLG motif [121,123]. It has subsequently been recognised that amino acids 23–36 around the DLG motif and amino acids 78 and 83 flanking the ETGE motif also contribute to the interaction with the Kelch-repeat domain of KEAP1 [124]. The existence of a low-affinity and a high-affinity binding site in NRF2 for KEAP1 results in a ‘two-site substrate recognition’ or ‘hinge and latch’ mechanism where the simultaneous docking of NRF2 onto the KEAP1 dimer through both of its DLG and ETGE motifs is required for repression of the CNC-bZIP transcription factor by KEAP1.

Following discovery of KEAP1, various hypotheses were proposed about how it might regulate NRF2 with cytoplasmic sequestration and phosphorylation allowing release of the transcription factor from KEAP1 and nuclear translocation, which has attracted much attention [reviewed in [125]]. These were, however, largely abandoned when it was recognised that KEAP1 is responsible for targeting NRF2 for rapid proteasomal degradation and that this ability is impaired by chemopreventive agents such as SFN [126]. Subsequently, it was demonstrated by various research groups that KEAP1 acts as a substrate adaptor for the CUL3 E3 ubiquitin ligase complex, called CRL^KEAP1^ [127,128,129,130]. CUL3 is a member of a conserved family that serve as scaffold proteins and allow the formation of a multi-subunit complex that contains an E2 ubiquitin-conjugating enzyme [131]. As shown in Figure 3, CUL3 contains three cullin repeat (CR1, CR2 and CR3) domains, a cullin homology (CH) domain, and a C-terminal neddylation site.

The prevailing view is that both the DLG and ETGE motifs of NRF2 must engage with the two Kelch-repeat domains in dimeric KEAP1 for the CNC-bZIP transcription factor to be ubiquitylated, and that chemopreventive agents and oxidative stressors readily perturb binding of NRF2 to KEAP1 by the low-affinity DLG motif, and so prevent both of the two-site interactions from being formed correctly. This, in turn, causes a logjam in the ubiquitylation pathway that allows newly translated NRF2 to bypass the CRL^KEAP1^ complex and translocate unimpeded to the nucleus where it dimerises with small MAF proteins to transactivate ARE-driven genes [132,133].

## 6. The Triggering of Multiple Thiol-Based Sensors in KEAP1 by Thiol-Reactive Small Molecules Results in De-repression of NRF2 and Protection against Oxidative Stress

Mouse and human KEAP1 comprise 624 amino acids, and amongst these residues the murine and human proteins contain 25 and 27 cysteine (Cys) residues, respectively. Recognition that roughly half of the Cys residues in mammalian KEAP1 proteins are positioned in close proximity to basic amino acids, suggested to a number of researchers that the -SH group of such residues would exist in a thiolate anion form (-S^−^) and so would react readily with soft electrophiles, making KEAP1 a strong candidate as the sensor of chemopreventive agents that stimulate NRF2-mediated gene induction. According to this proposal, it was envisioned that modification of certain Cys residues in KEAP1 would prevent it from allowing CRL^KEAP1^-directed ubiquitylation of NRF2. Once modified, KEAP1 is likely to be cleared from the cell by autophagy [134], and presumably any associated NRF2 that is bound just via its ETGE motif will be degraded in the same way [135].

Consistent with the idea that chemopreventive agents act as thiol-active reagents, Cys257, Cys273, Cys288 and Cys297 in purified recombinant mouse Keap1 were found to be modified by the inducer dexamethasone 21-mesylate [136]. Confirmation that Cys273 and Cys288 in Keap1 are required to suppress ARE-driven luciferase activity ex vivo was obtained by ectopic expression of wildtype and mutant forms of the adaptor protein in Keap1-ko embryonic fibroblasts [137]. Besides the requirement for Cys273 and Cys288 in KEAP1 to allow repression of NRF2, Cys151 in KEAP1 was found to be essential for the inducer SFN and the BHA metabolite *tert-*butyl hydroquinone (tBHQ) to block ubiquitylation of NRF2 by CUL3 [129], and residues Cys226 and Cys613 were found to be necessary for activation of NRF2 by Cd^2+^, As^3+^, Se^4+^ and Zn^2+^ [138] and by H_2_S [139]. More recently, it has been reported that Cys226, Cys613, Cys622 and Cys624 are engaged in the sensing of H_2_O_2_, through the formation of disulfide bonds between these residues [140]. Interestingly, there is redundancy in the system insofar as mutation of one of these four Cys residues still allows the mutant KEAP1 to sense H_2_O_2_ [140]. In view of the facts that the inherent reactivity of the Cys thiols in PRDX1 and PRDX2 are much higher than those in most proteins and that these enzymes are very abundant in the cell [141], an important possibility is that these thiol peroxidases act as the initial receptor for H_2_O_2_ and, following the formation of a disulfide bond within PRDX1 or PRDX2, relay this information to KEAP1 via the formation of mixed disulphide bonds (Figure 4). This would proceed via the formation of a disulfide exchange intermediate between PRDX1/2 and KEAP1, that would lead to PRDX1 and/or PRDX2 being reduced and KEAP1 oxidised, with the oxidised KEAP1 being ultimately reduced by the TXN1/TXNRD1 couple. Notably, this type of PRDX1/2-based redox relay system has been reported for apoptosis signalling kinase 1 (ASK1) and signal transducer, activator of transcription 3 (STAT3) and DJ-1 [142,143,144], and may also hold true for KEAP1. Whilst KEAP1 is widely recognised to contain reactive Cys residues, it is substantially less abundant than PRDX1 or PRDX2, and so how a H_2_O_2_ redox signal penetrates the antioxidant defence to be perceived by KEAP1 is unclear.

From research stimulated by an interest in chemoprevention, a picture has emerged of KEAP1 as a ubiquitin ligase substrate adaptor that allows cellular adaptation to a variety of redox stressors through complex thiol-based modifications to its various reactive Cys residues that alter its ability to repress NRF2. An interesting feature to emerge is that the Cys-based sensors in KEAP1 recognise different electrophilic agents, but this remains incompletely understood. Just how KEAP1 cross-talks with other components of redox signalling pathways, such as PRDXs, is not understood.

## 7. NRF2 is Frequently Upregulated in Malignant Disease

Over many years, studies of expression of detoxification enzymes in preneoplastic rat hepatocyte nodules, human tumours and drug-resistant cancer cell lines have provided compelling evidence that GSH, glutathione-biosynthetic enzymes, GSTs, UGTs, NQO1, EPHX1 and AKRs are frequently overexpressed in malignant cells and that this is often associated with resistance to chemotherapeutic agents [71,80,145,146,147,148,149]. With the benefit of hindsight, these findings should have led researchers to anticipate that NRF2 was likely to be commonly upregulated during tumourigenesis. However, the literature indicating that NRF2 mediates chemoprevention seemed at odds with such an interpretation. Evidence that repression of NRF2 by KEAP1 is impaired in human cancer came initially from the laboratory of Shyam Biswal through the demonstration that the *KEAP1* gene is subject to somatic mutations in some cell lines and clinical samples from patients with non-small cell lung cancer (NSCLC) [150]. In particular, these researchers identified mutations in *KEAP1* that occurred throughout regions encoding the BTB, IVR and Kelch-repeat domains, findings that were confirmed and extended by others [151]. Shortly thereafter, it was discovered that somatic mutations in *NFE2L2* commonly occur in NSCLC and oral cancer cell lines as well as various primary lung cancers and primary head and neck cancers, and that the somatic mutations resulted in amino acid substitutions at residues associated with the DLG and ETGE motifs in NRF2 [152].

Examination of second-generation sequencing of a large number of tumours deposited in the catalogue of somatic mutations in cancer (COSMIC) and the cancer genome atlas (TCGA) databases revealed that *NFE2L2* and *KEAP1* are mutated in various common cancers, including lung, head and neck, and bladder [153,154,155,156,157,158,159] (see Table 2). It has also been noted that intragenic deletions occur in *NFE2L2* resulting in translation of NRF2 isoforms that lack amino acids that interact with KEAP1 in patients with squamous NSCLC and head and neck carcinoma [160]. Based on the facts that the somatic mutations in *NFE2L2* are restricted to regions of the gene encoding the DLG and ETGE motifs and their flanking amino acids, that such ‘hot spot’ mutations are presumed to result in gain-of-function in NRF2, and that *NFE2L2* is sometimes amplified, NRF2 has been designated an oncogene [161]. Using a variety of computational bioinformatic tools *NFE2L2* mutations have been considered likely to drive lung, oesophageal, cervical, bladder and uterine cancers, though this has not been formally tested [162,163]. As shown in Figure 5, NRF2 can also be upregulated by oncogene-stimulated transactivation of *NFE2L2*, as demonstrated for Kras, Braf and Myc [164].

As somatic mutations in *KEAP1* occur across the entire gene and cause loss-of-function, it has been referred to as a tumour suppressor [161], and it has been proposed to drive cancers of the lung, head and neck, and liver [162,163]. Besides somatic mutations, *KEAP1* is also sometimes subject to intragenic deletions or its expression suppressed by hypermethylation or by micro-RNAs [167,168,169]. Loss of SWI/SNF chromatin-remodeling complex activity also increases NRF2 activity [170]. Moreover, the ability of KEAP1 to repress NRF2 can be inhibited by oncometabolites such as fumarate [171,172], methylglyoxal [173] and succinylacetone [174], or NRF2 can be upregulated by overexpression of proteins that contain ETGE, or ETGE-like, motifs resulting in NRF2 being out-competed for binding to KEAP1; examples include iASPP, CHD6, DPP3, FAM129B, IKKβ, LAMA1, MCM3, SQSTM1/p62, PALB2, PGAM5 and WTX [175].

Although no mutations in *KEAP1* or *NFE2L2* have been reported in tumours of the central nervous system [155], NRF2 is hyperactivated in a subset of glioma patients, whose tumours display a mesenchymal subtype [176]. Evidence suggests a positive feedback loop between p62/SQSTM1 and NRF2 is responsible for NRF2 upregulation. Of interest, NRF2 has also been shown to facilitate glioma development by inducing expression of the transcriptional co-activator TAZ, a component of the Hippo signalling pathway [177]. Upregulation of NRF2 in glioma cells accelerates proliferation and oncogenic transformation, and protects against ferroptosis [178]. Additionally, IDH1-mutant glioma cells are dependent on NRF2-target genes to scavenge ROS [179].

## 8. Constitutive Activation of NRF2 Supports Post-Initiation Stages of Cancer

Whilst the designations of NRF2 and KEAP1 as an oncoprotein and a tumour suppressor protein, respectively, seem appropriate they should be treated with caution as there is no evidence that chronic activation of NRF2 is sufficient to initiate tumourigenesis. Rather, on its own, chronic activation of NRF2 can sometimes result in hyperplasia, but not cancer. Thus, modest permanent activation of Nrf2, as observed in the floxed *Keap1^FA/FA^* hypomorphic mouse, in which Keap1 is effectively constitutively knocked down, provides increased cytoprotection and has not been reported to result in any obvious long-term susceptibility to tumourigenesis [180]. Global knockout of Keap1 in the mouse, which provides higher upregulation of Nrf2 than observed in *Keap1^FA/FA^* mice, was by contrast, found to result in severe hyperkeratosis of the oesophagus and forestomach that resulted in postnatal death from malnutrition within 3 weeks of age, but the *Keap1^−/−^* mice showed no signs of malignant disease [181]. Moreover, liver-specific knockout of Keap1 (e.g., *Keap1^FA/FA^*; *Alb-Cre* mice) has not been reported to result in hepatocellular carcinoma, despite the animals having been first reported in 2006 [182]. Furthermore, persistent hyperactivation of NRF2 in human A549 lung cancer cells results in enhancer remodelling, allowing transcriptional activation of *NOTCH3* by an NRF2-CEBPB complex that strongly supports tumourigenesis, and which is not observed in normal cells under stress conditions, indicating that continual activation of NRF2 in cancer cells results in overexpression of an enlarged battery of genes that is distinct from that induced by NRF2 in normal cells [183].

A number of researchers have studied the effect of expressing mutant hyperactive forms of NRF2, modelled on somatic mutants of *NFE2L2* identified in clinical lung cancer samples (e.g., Nrf2^E79Q^ and Nrf2**^Δ^**^Neh2^), and none of these have provided evidence that chronic Nrf2 hyperactivity is sufficient to initiate tumourigenesis. Thus, generation of a *Nfe2l2^LSL-E79Q^; Krt14-Cre* mouse in which Nrf2^E79Q^ is expressed in keratin 14-positive tissues, revealed that expression of a constitutively active mutant form of the CNC-bZIP transcription factor resulted in hyperplasia of squamous cell tissue of the tongue, oesophagus and forestomach, but not squamous cell carcinoma, and that in oesophageal tissue this was associated with overexpression of well-recognised members of the ARE-gene battery, as well as genes for growth factors and related factors such as those encoding Areg, Bmp6, Epgn, Ereg, Hbegf, Myc, Vegfa and Wnt5a [184]. Similarly, hepatocyte-specific expression of Nrf2^E79Q^ resulted in hepatomegaly that was accompanied by overexpression of *Areg* and *Tgfa*, which encode ligands for the EGFR, and *Pdgfc*, which encodes a ligand for the PDGFR, but did not result in liver cancer [185]. In a mouse model in which a constitutively active form of Nrf2 that lacks amino acids 1–88 (i.e., Nrf2**^Δ^**^Neh2^, called caNrf2) was expressed from the *keratin 5* promoter (with the line called K5cre-caNrf2), the mutant mice developed chloracne-like skin disease with hyperkeratosis of hair follicles in the epidermis [186], but not cancer. However, when used in skin carcinogenesis experiments, the K5cre-caNrf2 mouse was found to support survival of keratinocytes that harboured oncogenic mutations. In these experiments, crossing K5cre-caNrf2 mice with a HPV8-stimulated skin tumourigenesis mouse produced better survival of oncogene-expressing cells than when a DMBA/TPA two-step chemical carcinogenesis model was employed [187]. By contrast with these findings, it has been reported that ectopic expression of caNrf2 in primary breast cancer cells had no significant effect on the rate of primary tumour formation [188]. Taken together, these different mouse models all suggest that constitutive activation of Nrf2 is not sufficient to initiate cancer but may support cell growth, though this likely depends on the oncogene(s) involved.

Expression of Nrf2**^Δ^**^Neh2^ in mouse fibroblasts (i.e., in Colα2Cre-caNrf2 mice) was found to elicit a senescence-associated secretory phenotype with increased expression of cytokines, growth factors and extracellular matrix proteins that affected neighbouring cells, suggesting it conferred a cancer-associated fibroblast phenotype [189]. This finding seems at variance with other reports that activation of NRF2 delays senescence in human fibroblasts [190], and may simply reflect the hyperactive nature of Nrf2**^Δ^**^Neh2^ or other context-dependent factors. Interestingly, through use of both floxed *Keap1^FA/FA^* hypomorphic mice and floxed *Keap1^FB/FB^* “normal” mice in a *Kras^LSL-G12D/+^* mouse lung cancer model, Masi Yamamoto and colleagues showed that activation of Nrf2 in the microenvironment surrounding the tumour restricts progression of lung cancers in which Nrf2 is upregulated, and bone marrow transplantation experiments revealed that activation of the transcription factor in hematopoietic cells contributed significantly to suppression of the tumour [191], findings that suggest high NRF2 activity can suppress the later stages of cancer by supporting immune cell function.

In conclusion, based on studies of the evolution of NSCLC, it seems likely that NRF2 is upregulated at an early adenocarcinoma stage of disease [192], which accords with data from rat liver preneoplastic nodules obtained using the Solt-Farber protocol [193,194]. Collectively, these results suggest that NRF2 upregulation is not a potent cancer driver, and that alone, its constitutive activation cannot initiate tumourigenesis. Provocatively, upregulation of NRF2 in immune cells seems to combat late progression and dissemination of malignant disease.

## 9. Mechanisms by Which Upregulation of NRF2 Supports Post-Initiation Stages of Cancer

Given that hyperactivation of NRF2 is not sufficient to initiate tumourigenesis, it is necessary to establish why it is upregulated in certain types of cancer. The answer appears to be that NRF2 hyperactivation contributes to tumourigenesis by ameliorating oxidative stress, supporting cell growth/proliferation by various means (e.g., increasing the PPP, serine synthesis and autophagy) and attenuating the immune system. Before addressing the question of why NRF2 is upregulated in certain cancers, it should first be asked whether NRF2 is required for carcinogenesis under conditions where it is subject to normal homeostatic control. Research into the inhibitory effects of oxidative stress on tumourigenesis in a murine Kras oncogenic pancreatic cancer model has revealed that Nrf2 is necessary for Kras-driven pre-invasive pancreatic intraepithelial neoplasia as the number of lesions in *Kras^LSL-G12D/+^; Nfe2l2^−/−^* mice was much lower than in *Kras^LSL-G12D/+^; Nfe2l2^+/+^* mice [164]. Moreover, using *Alb-Cre; Kras^LSL-G12D/+^; p53^LSL-R172H/+^* mice, deletion of *Keap1* has shown that upregulation of Nrf2 accelerated Kras/p53-driven cholangiocarcinoma [195]. Similarly, using a CRISPR-Cas9 strategy, deletion of *Keap1* has been found to accelerate Kras-driven lung adenocarcinoma [196]. Additionally, constitutive activation of Nrf2 has been shown to promote lung cancer in a mouse model initiated by knockout of phosphatase and tensin homolog (Pten) by administering adenoviral Cre to *Keap1^FB/FB^; Pten^F/F^* mice, with the latter mutation causing permanent activation of protein kinase B (PKB)/Akt [197].

As cancer cells produce high levels of ROS to sustain proliferation, and also have to withstand oxidative stress during metastasis, it seems likely that upregulation of Nrf2 benefits the tumour because the resulting overexpression of antioxidant genes prevents ROS-stimulated cell death (reviewed in [8]). Besides directing the expression of genes encoding enzymes that scavenge ROS, NRF2 also controls the expression of PPP genes in both the oxidative (*G6PD* and *PGD*) and non-oxidative (*TALDO1* and *TKT*) arms, and overexpression of these may contribute to survival and proliferation of cancer cells by increasing synthesis of NADPH and ribonucleotides [198]. Of note, rapidly proliferating cancer cells primarily use the non-oxidative arm of the PPP to generate ribonucleotides for nucleic acid biosynthesis [198].

In experimental cancer models, the mechanisms by which Nrf2 upregulation promotes carcinogenesis have not been rigorously proven, at least they have seldom entailed knockout of individual Nrf2-target genes. An example of this approach is the tamoxifen-induced conditional knockout of *Slc7a11* in pancreatic ductal adenocarcinoma cells generated in *Kras^LSL-G12D/+^*; *Tp53^R172H/+^*; *Pdx1FlpO^tg/+^*; *Slc7a11^Fl/Fl^*; *Rosa26^CreERT2/+^* mice, experiments in which knockout of *Slc7a11* in the tumour resulted in ferroptotic death of the tumour cells and increased survival [199]. Whilst the above study focused on the importance for tumour survival of maintaining intracellular cysteine levels to combat ROS-stimulated death, work with the murine *MMTV-PyMT* spontaneous mammary cancer model, crossed onto a *Gclm^−/−^* background, along with use of the GCL inhibitor buthionine sulfoximine and the TXNRD inhibitor auranofin, indicated that the GSH antioxidant system supports early tumourigenesis and that the TXN system can compensate for depletion of GSH by increasing cystine import via Slc7a11, inferring a level of redundancy [200]. As Nrf2 controls both GSH- and TXN1-based antioxidant systems, it is likely a major factor in determining whether cancer cells can withstand oncogene-generated ROS.

In a model of recurrent breast cancer, tumour cells that survived oxidative stress caused by oncogene inhibition did so by NRF2-directed metabolic reprogramming that entailed overexpression of genes for oxidative PPP enzymes and TXN1, TXN2 and TXNRD1, but not those for GSH synthesis, suggesting NRF2 aids formation and growth of recurrent tumours [188]. Importantly, in addition to NRF2-directed overexpression of antioxidant enzymes, other mechanisms can be utilized to increase GSH during the later stages of cancer development including, for example, increased generation of NADPH via the folate pathway during metastasis [201] and activation of mTOR signalling as well as increased mitochondrial metabolism and glutamine flux stimulated by estrogen-related receptor α (ERRα) in lapatinib-resistant breast cancer cells [202].

The fact NRF2 regulates metabolic flux through the PPP suggests it is likely to aid tumour survival and growth by directing NADPH generation and de novo ribonucleotide synthesis [31]. Treatment of *Kras^LSL-G12D/+^; Keap1^FB/FB^* mice (that had already received adenoviral Cre intranasally) with the PGD inhibitor 6-aminonicotinamide prevented formation of lung adenomas and resulted in just lung hyperplasia [203], findings that suggest the PPP contributes substantially to the ability of upregulated Nrf2 to support tumourigenesis, though the extent to which this is due to increased production of NADPH or ribonucleotides requires clarification. Moreover, as NRF2 regulates ATF4, it indirectly controls serine synthesis by regulating expression of genes encoding phosphoglycerate dehydrogenase (PHGDH), phosphoserine aminotransferase-1 (PSAT1) and serine hydroxymethyltransferase-2 (SHMT2), and as serine is used as an intermediate in glutathione and nucleotide production, NRF2 upregulation supports cancer cell growth and proliferation [204].

Besides antioxidant systems and the PPP, NRF2 also regulates expression of proteasome subunits and components of the autophagy system, and overexpression of these upon upregulation of NRF2 likely also contributes to cancer cell survival and proliferation [205]. Furthermore, autophagy plays a crucial role in inhibiting tumourigenesis in the liver, presumably by preventing accumulation of dysfunctional mitochondria and ameliorating oxidative stress, as evidenced by the spontaneous development of hepatic adenomas in mice with mosaic deletion of Atg5 and hepatocyte-specific deletion of Atg7 [206]. Once tumours are established however, autophagy often promotes cancer cell growth by recycling non-essential cellular components to support oxidative phosphorylation and thereby overcome nutrient stress [207,208]. Intriguingly, inhibition of autophagy, by chloroquine treatment or knockout of *ATG7*, in autophagy-dependent cancer cells stimulates adaptation by NRF2-mediated induction of proteasomal subunit genes, with largest increases observed for *PSMB8, PSMB9, PSMB10* and *PSMC1*, indicating NRF2-mediated regulatory crosstalk between the proteasome and autophagy [36].

Obesity-stimulated liver steatosis represents a growing cause of hepatocellular carcinoma, and is associated with inhibition of autophagy and the accumulation of p62/SQSTM1 [209]. In the liver, phosphorylation of LC3 (i.e., microtubule-associated proteins 1A/1B light chain 3B) at Ser12 by the atypical protein kinase C (PKC)λ/ι inhibits autophagy by preventing LC3 from interacting with p62/SQSTM1 [210]. In mice treated with DEN and fed a high-fat diet, hepatocyte-specific knockout of PKCλ/ι (in *Alb-Cre; Prkci^F/F^* mice) resulted in multiple hepatocellular carcinomas, some of which were aggressive, whilst similarly treated *Prkci^F/F^* mice only developed benign hepatic adenomas [210]. These workers found knockout of PKCλ/ι increased autophagy and ROS levels, causing oxidative stress and, consequently, activation of Nrf2. They also found hepatocytes from *Alb-Cre; Prkci^F/F^* mice exhibited higher rates of proliferation than controls, and that knockdown of Nrf2 suppressed proliferation in the PKCλ/ι-knockout hepatocytes. Together, these results indicate Nrf2 is an important component of the tumour environment when autophagy is increased but they do not give an insight into whether NRF2-mediated overexpression of autophagy genes contributes to tumourigenesis.

As NRF2 regulates ATF4, it is likely to indirectly control the expression of cyclooxygenase 2 (COX2) and production of prostaglandin E_2_ (PGE_2_), which suggests NRF2 upregulation may foster a microenvironment that allows tumour cells to evade the innate immune system. In melanoma cells, knockout of NRF2 has been shown to decrease COX2 levels and PGE_2_ production, and this was associated with a large increase in expression of innate immune response genes involved in defence against viral infection, such as *Rsad2*, *Ifih1*, *Ifit1* and *Isg15* [211]. Whilst these findings suggest NRF2 upregulation can suppress innate immune responses, this conclusion should be tempered by the finding that activation of Nrf2 in the microenvironment surrounding the tumour restricts progression of lung cancers [191].

Lastly, it is well recognised that NRF2 upregulation is associated with chemoresistance and radioresistance, due to overexpression of drug-metabolising enzymes and antioxidant systems [205]. The overexpression of drug-metabolising genes caused by hyperactivation of NRF2 is likely to increase resistance to ferroptosis because AKR isoenzymes catalytically reduce lipid peroxidation products that trigger ferroptotic cell death [212]. Of note, AKR1C1, AKR1C2 and AKR1C3 are particularly effective at reducing 4-hydroxy-2-nonenal to 1,4-dihydroxy-2-nonene [213].

## 10. Evidence of Dysregulation of NRF2 in Human Cancers and Segregation with Activated Oncogenes

### 10.1. Upregulation of NRF2 in Lung Tumours

Lung cancer is the leading cause of cancer associated mortality world-wide [214], partly due to its highly heterogenous nature and often late stage of diagnosis. Environmental factors such as exposure to cigarette smoke, asbestos, radon gas, arsenic and silica have been linked to lung cancer development and poor outcomes [215,216]. For example, smokers are 10–20 times more likely to develop the disease than non-smokers [215]. NSCLC is the major histological subtype of lung cancer, representing >80% of all lung cancer cases [216,217]. NSCLC can be further histologically subdivided, according to tissue of origin, into adenocarcinoma (LUAD, which is the most prevalent), squamous cell carcinoma (LUSC) and large cell carcinoma [218]. Molecular profiling of both LUAD and LUSC revealed thirty-eight genes frequently mutated in LUAD and twenty frequently mutated genes in LUSC, with six commonly mutated genes in both lung cancer subtypes: these are, *TP53*, *RB1*, *ARID1A*, *CDKN2A*, *PIK3CA* and *NF1* [153] (see Table 3 for the frequency of mutations in genes implicated in NRF2 activation in NSCLC). Due to the few overlapping mutations, it is thought that somatic mutations in driver genes differ in LUAD and LUSC.

Large-scale molecular profiling by the Cancer Genome Atlas Research Network of 230 LUAD samples from untreated patients revealed eighteen frequently mutated genes, including *EGFR, KRAS, STK11* and *KEAP1* [219]. When pathway analysis was carried out, alterations in those linked to oxidative stress were found in 22% of tumours due to mutations in *KEAP1* (mutated in 19%), *CUL3* (mutated in >1%) and *NFE2L2* (mutated in 3%) [219]. Amongst 178 LUSC samples from untreated patients, ten genes were found to be frequently mutated, including *TP53*, *CDKN2A*, *PTEN*, *PIK3CA*, *KEAP1* and *NFE2L2* [220]. These data also revealed that LUSC have relatively few mutations in *EGFR* and *KRAS*, which are commonly mutated in LUAD, but a relatively high frequency of mutations in *PTEN* and *PIK3CA.* However, they highlighted that dysregulation in oxidative stress pathways, as seen in the LUAD cases, was similarly observed in 34% of LUSC tumours, due to mutations (or copy number alterations) in *KEAP1* (mutated in 12%), *CUL3* (mutated in 7%) and *NFE2L2* (mutated in 19%). The study also highlighted that mutations in *KEAP1* and *CUL3* are often loss-of-function mutations that occur mutually exclusive to mutations in *NFE2L2* [220]. Somatic mutations in *NFE2L2* were found to be clustered around those parts of the gene encoding the DLG and ETGE motifs, with this type of ‘hot spot’ mutational pattern considered characteristic of an oncogene [221]. Unlike mutations in *NFE2L2*, somatic mutations in *KEAP1* were found to be distributed throughout the gene, which represents the mutational pattern associated with tumour suppressor genes [151,222]. Due to their wide distribution, it is difficult to attribute a function to all the mutations in *KEAP1*; however, some have been studied such as the R32Q mutant, which has been shown to drive carcinogenesis [217]. The majority of mutations in *KEAP1* do not affect the binding and ubiquitylation of NRF2 but somehow halt proteasomal degradation of the transcription factor [222].

It is now recognised that mutations in *KEAP1*, *NFE2L2*, *PTEN* and *EGFR* rarely co-exist. Mutations in *NFE2L2* have been demonstrated to co-occur with *PIK3CA* mutations [223] and mutations in *TP53*. By contrast, mutations in *KEAP1* tend to co-occur with mutations in *KRAS* and *STK11* [203]. This pattern of mutual exclusivity of some mutated genes with others is indicative of a level of redundancy, with both mutations resulting in the same phenotype (see Table 4, Table 5 and Table 6 for co-occurrence of mutations). Another point of divergence between *KEAP1* and *NFE2L2* mutations, is that those in *KEAP1* are more commonly associated with LUAD than LUSC [224]. Research has also revealed that several mutations in *KEAP1* are G>T transversions, which are characteristic mutations associated with exposure to tobacco smoke, and may in part explain why *KEAP1* mutations often co-occur with *KRAS* mutations, which are also associated with smoking [150,196,225]. The co-occurrence of *KEAP1* and *KRAS* mutations suggests that *KEAP1* mutations either affect NRF2 activity differently than KRAS-stimulated overexpression of NRF2, or are affecting other signalling pathways that are beneficial to the tumour.

By comparison with mutations in *KEAP1* and *NFE2L2*, *CUL3* mutations have not attracted much attention. Mutations in *CUL3* co-occur with mutations in *NFE2L2* in LUSC, suggesting that a mutation in *CUL3* might potentially influence other cellular functions besides those controlled by NRF2 [226]. CUL3 is a member of the cullin family of scaffold proteins that bind E2 ubiquitin-conjugating enzymes and recruit protein substrates for ubiquitylation [227]. Whilst CUL3 binds the BTB domain of KEAP1, there are over 200 BTB domain-containing proteins in humans, all of which may be bound by CUL3 [226,228]. Therefore, some of the mutations in *CUL3* may not affect NRF2 degradation but that of oncogenic proteins, implying that mutations in *CUL3* may enhance oncogenesis independently of NRF2. Also, the wide range of potential CUL3-binding partners, which will affect a wide range of signalling pathways, may explain the lower frequency of *CUL3* mutations than *NFE2L2* or *KEAP1* mutations [226].

Targeted therapies for patients with mutations in specific genes, such as *EGFR* and *BRAF*, have had very promising results in LUAD [219], but due to the genetic diversity between the different histotypes of NSCLC, such therapies are not applicable for LUSC [220]. One caveat of the analyses carried out by TCGA is that samples analysed were collected from patients with early-stage disease [229]. Research has now shown that mutations in *KEAP1* and *NFE2L2* are often associated with late-stage metastatic disease [230], with stage-4 cancer patients showing higher NRF2 activity than stage-3 patients [231]. When comparing the impact of mutations in *NFE2L2* or *KEAP1* on expression of NRF2-target genes, mutations in *NFE2L2* were found to result in overexpression of more genes compared to mutations in *KEAP1*, suggesting that amino acid substitutions within the transcription factor are a more robust way for the tumour to command a comprehensive oxidative stress response [218].

Since mutations in *KEAP1* and *NFE2L2* do not co-exist, there appears to be a segregation of each with regard to how NRF2 upregulation enhances or supports different driver mutations. In view of the fact that NRF2 upregulation is not associated with cancer initiation, but with tumour progression, it would be reasonable to suggest that the mutation in *KRAS,* or *TP53,* arises as an early event that precedes mutation in *KEAP1,* or mutation in *NFE2L2*, respectively. This has been highlighted by the findings that *KRAS* mutant tumours with co-occurring *KEAP1* mutations are associated with later stage lung cancer [203].

### 10.2. Upregulation of NRF2 in Oesophageal Tumours

Oesophageal cancer is the 7th most common form of malignant disease worldwide [214]. It can be divided epidemiologically into two subtypes, namely adenocarcinoma (EAC) and squamous cell carcinoma (ESCC). ESCC is the more prevalent, and is associated with environmental exposure to alcohol and cigarette smoke, implicating a strong contribution by ROS to the disease [232]. ESCC patients receive chemoradiotherapy in the form of radiation combined with 5-fluorouracil or a platinum-based agent [233]. The genetic components of ESCC have been extensively characterized. Next-generation sequencing and whole exome sequencing studies have revealed mutations in *TP53* (most mutated gene in ESCC), *NOTCH1*, *PIK3CA*, *TGFBR2* and *NFE2L2* [232,234].

In ESCC, the somatic mutations in *NFE2L2* cluster to regions of the gene encoding amino acids in NRF2 associated with the KEAP1-binding sites, similar to mutations found in LUSC. Interestingly, unlike other cancers with high NRF2 activity, *KEAP1* mutations rarely occur in ESCC. Also, mutations in *CUL3* were rarely observed [154,235]. *NFE2L2* mutations co-occur with mutations in *TP53* and *NOTCH1*, but are mutually exclusive with mutations in *PIK3CA* [235]. However, due to their low frequency in ESCC, we cannot assess the co-occurrence of *KEAP1* and *CUL3* mutations with mutations in other genes.

NRF2 activity is commonly upregulated in ESCC tumours and cells lines [236]. This high incidence in NRF2 upregulation cannot be attributed solely to mutations in *NFE2L2, KEAP1* and *CUL3*, nor can it be attributed to mutation of oncogenic driver genes *KRAS*, *BRAF* and *MYC* that increase *NFE2L2* expression [154,232,235]. Rather, this unexpectedly high incidence of NRF2 upregulation may be due to altered expression of microRNAs (*miRs*) that target *NFE2L2* and *KEAP1* expression; these are small endogenous non-coding RNAs that can regulate gene expression by altering translation or stability of a target through directly binding to the 3′-untranslated region (UTR) or coding region, and can, therefore, regulate gene expression and tumour progression by functioning as tumour suppressors/oncogenes [237]. Several *miRs* that exhibit altered expression in ESCC are able to regulate the activity of either NRF2 or KEAP1, and may provide an alternative route by which the expression of NRF2-target genes can be enhanced in ESCC tumours lacking *NFE2L2* mutations. For example, *miR-200a* directly targets *KEAP1*, increases NRF2 protein abundance and NRF2 nuclear accumulation in ESCC cells [238]. The tumour suppressor *miR-153-3p* directly binds to the 3′ UTR of NRF2 inhibiting its expression [237], and *miR-432-3p* downregulates *KEAP1* expression by directly targeting the coding region [168].

Elevated NRF2 abundance and expression of the NRF2-target genes *GCLC* and *NQO1* have been found in both ESCC tumours and cell lines, and are associated with poor patient survival [239,240]. The exact timing at which NRF2 activity is elevated in oesophageal carcinogenesis has not been determined but analysis of in situ ESCC tumours revealed the absence of *NFE2L2* mutation, suggesting that these mutations may not be associated with the initial stages of ESCC cancer but may occur later to influence cancer progression [239].

### 10.3. Upregulation of NRF2 in Liver Tumours

The liver is the primary site of detoxification in the human body and is a metabolically active organ that is exposed to substantial levels of oxidative stress [241]. Hepatocellular carcinoma (HCC) is the 6^th^ most common cause of cancer associated mortality worldwide [214]. There are several well-documented risk factors associated with the development of HCC, these include exposure to viral hepatitis B or C, high alcohol intake, diabetes, non-alcoholic fatty liver disease and liver cirrhosis, which contribute to the development of 80–90% of all HCC [242,243]. Analysis of somatic mutations present in HCC revealed a high frequency of mutations in *TERT* (most commonly mutated gene), *CTNNB1*, *TP53* and *ARID1A* [244].

Mutations in *NFE2L2*, that specifically cluster to regions of the gene encoding amino acids in NRF2 that bind KEAP1, have been documented in early preneoplastic liver lesions and HCC tumour samples [245]. *KEAP1* mutations also occur in HCC tumours and when mapped, are found throughout the gene [155]. Interestingly, the mutational frequency in *NFE2L2* and *KEAP1*, 3% and 5% respectively, is much lower than in some other cancer types [244]. Also, *CUL3* mutations were not found in the datasets analysed. *NFE2L2* mutations and *KEAP1* mutations were found to both co-occur with mutations in *TP53*, *CTNNB1* and *ARID1A*, but not with mutations in *TERT* [155,246].

Autophagy has been proposed to play a key role in protecting the liver against disease by preventing the accumulation of damaged cytoplasmic proteins and organelles [247]. It is thought to protect against the initiation of carcinogenesis and is frequently impaired during HCC, leading to diminished clearance of cellular constituents. Inappropriate activation of the Hippo signalling pathway effector Yes associated protein 1 (YAP1), which acts as a transcriptional coactivator of TEA domain (TEAD) family members 1–4 (TEAD1-4) and so influences expression of genes that control cell survival, proliferation and polarity, occurs as an early event during the development of HCC [248]. The activation of YAP1 during early HCC development is thought to be due to its reduced degradation by autophagy [249]. Besides YAP1, autophagy positively controls NRF2 by virtue of the fact that the autophagy cargo receptor p62/SQSTM1 binds KEAP1 and by suppressing KEAP1 increases NRF2 activity [134]. It is now recognised that under conditions of oxidative stress, p62/SQSTM1 is phosphorylated at Ser351 by either TBK1 or CK2, which enhances binding between KEAP1 and p62/SQSTM1, preventing KEAP1 binding to NRF2, leading to NRF2 nuclear accumulation and induction of ARE-driven genes [250]. From these findings, it might be anticipated that loss of autophagy during early stages of HCC would result in accumulation of KEAP1 and downregulation of NRF2, but this does not seem to be the case. Presumably, loss of autophagy results in an increase in p62/SQSTM1 as well as KEAP1, which blunts the ability of KEAP1 to target NRF2 for proteasomal degradation. Indeed, the interaction between p62/SQSTM1 and KEAP1 is thought to provide HCC cells with protection against ferroptosis by increasing the expression of members of the ARE-gene battery that are critical to iron and ROS metabolism [251]. Also, of relevance, is that in instances of HCC that involve hepatomegaly, NRF2 supports liver growth that is associated with enhanced glycogenosis and PKB/Akt signalling [185]. Further work is required to translate the mouse-based experimental work on autophagy in HCC to the clinical setting.

High levels of oxidative stress have been implicated in the development of HCC, potentially through dysregulation of autophagy and upregulation of NRF2 [241]. The stage of cancer at which mutations in *NFE2L2* and *KEAP1* arise is currently unknown. However, it has been shown that mutations in *NFE2L2* occur in the early stages of HCC, and whilst the prevalence of *NFE2L2* mutations decreases as the disease increases in severity, NRF2 activity remains heightened throughout HCC progression [252]. Potentially, this is due to high levels of oxidative stress leading to phosphorylation of p62/SQSTM1, which allows it to compete for binding to KEAP1, resulting in increased NRF2 activity in the later stages of HCC. Whether there is any difference between NRF2 activation occurring through somatic mutations or through alternative routes is currently unknown.

### 10.4. Upregulation of NRF2 in Head and Neck Tumours

Head and neck squamous cell carcinoma (HNSCC) is one of the most common types of cancer worldwide, and has a poor associated survival [253]. HNSCC represents a heterogenous group of malignant diseases that includes cancer of the oral cavity, oropharynx, nasopharynx, hypopharynx, larynx, soft tissue of the neck, salivary glands and mucosal membranes [254]. Several etiological factors have been linked to HNSCC, including tobacco use, alcohol consumption and human papilloma virus (HPV) status [255]. Whole exome sequencing of HNSCC samples has shown a high frequency of mutations in *TP53*, *CDKN2A*, *FAT1* and *PIK3CA* [156].

*NFE2L2* and *CUL3* mutations have been identified in HNSCC at a frequency of 6% and 4%, respectively, with mutations in *NFE2L2* mapping onto regions encoding the Neh2 domain of NRF2 and mutations in *CUL3* occurring in regions encoding amino acids 33–66 in the CR1 domain. Mutations in *KEAP1* are rare and when they occur, are spread throughout the coding region [156].

Mutations in *NFE2L2* co-occur with mutations in all the commonly mutated genes found in HNSCC, whereas *CUL3* mutations co-occur with mutations in *TP53*, *PIK3CA* and *FAT1* but not *CDKN2A* [156]. Aside from somatic mutations, NRF2 may be upregulated in HNSCC due to the presence of copy number amplifications in *NFE2L2*, copy number deletions in *CUL3* and hypermethylation of *KEAP1* [156,256].

Proteomic profiling revealed an NRF2 gene signature in HNSCC that is associated with poor patient survival and increased NRF2 target gene expression, particularly HMOX1, GSTs and AKRs [254]. Although HNSCC encompasses a wide range of tumours that differ dramatically not only physically but genetically, additional diversity is provided by etiological factors such as HPV. In tumours of patients that are HPV-negative, high frequencies in mutation of *TP53* is observed, whereas the opposite is seen in tumours of HPV-positive patients [257]. As data suggest, *NFE2L2* mutations tend to co-occur with *TP53* mutations, this would suggest they are associated with HPV-negative tumours. Work by Victor Martinez and colleagues specifically focused on the consequences of NRF2 upregulation in HNSCC, and found that whilst somatic mutations in *NFE2L2* and *KEAP1* are rare events, they are associated with poor outcome [256]. Treatment failure in HNSCC patients has also been linked to the presence of cancer stem cells, which are a small population of cells that can rapidly proliferate, are resistant to apoptosis and exhibit increased NRF2 abundance [258].

### 10.5. Upregulation of NRF2 in Gastric Tumours

Gastric cancer (GC) is the 5th most common cause of cancer worldwide and more prevalent in the male population [214]. Risk factors associated with the development of GC include dietary factors such as salt consumption and vitamin A and vitamin C levels, chronic atrophic gastritis, and *Helicobacter pylori* (*H. pylori*) infection [102]. Oxidative stress has also been implicated in GC development by damaging DNA and causing mucosal injury. Studies into the genetics of GC have revealed a high frequency of mutations in *TP53* (most commonly mutated gene), *LRP1B*, *AIRD1A, PCLO* and *PIK3CA* [155,157].

Molecular profiling of stomach adenocarcinoma patient tissues has identified mutations in *KEAP1*, *NFE2L2* and *CUL3*, with *KEAP1* mutations being the most prevalent [155,157]. Interestingly, mutations in *NFE2L2* do not map to parts of the gene encoding the Neh2 domain of NRF2, and so might be passenger mutations. Somatic mutations in *KEAP1* and *CUL3* map to amino acids throughout the KEAP1 protein and to the CR1 domain of CUL3. Mutations in *KEAP1* and *CUL3* co-occur with mutations in *TP53* and *PIK3CA*. *NFE2L2* mutations are mutually exclusive with mutations in *TP53* but also co-occur with *PIK3CA* mutations, though it should be noted there were very few samples with *NFE2L2* mutations in these studies.

As increased nuclear levels of NRF2 protein have been documented in gastric cancer cell lines and tissues [259], but the frequency of somatic mutations in *NFE2L2*, *KEAP1* and *CUL3* is low, it is likely that NRF2 can be upregulated by other mechanisms in GC. Analyses by the Cancer Genome Atlas Research Network has revealed that GC patients with copy number amplifications in *NFE2L2* often harbour mutations in *TP53* and exhibit copy number amplification in *PIK3CA*. Deletions in *KEAP1* were also found in some GC patients and tended to co-occur alongside *TP53* mutations. High levels of NRF2 mRNA and protein have been found in GC tissue in comparison to control tissues and have been linked with clinicopathological features such as tumour size and drug resistance [259].

### 10.6. Upregulation of NRF2 in Bladder Tumours

Urinary bladder cancer (UBC) is responsible for a large number of cancer-associated deaths per year world-wide and is 4-times more common in men than women [260]. UBC can be broadly divided into two groups as follows: an early stage, low grade (commonly T1) non-invasive and more prevalent group called non-muscle invasive bladder cancer (NMIBC); a late stage, high grade (commonly T3/T4) invasive and more rare type called muscle invasive bladder cancer (MIBC) [261]. Risk factors for the development of UBC include cigarette smoke, arsenic exposure, inhalation of diesel fumes and aging [260,261]. Interestingly, molecular profiling has revealed that different genes are somatically mutated in NMIBC and MIBC. MIBCs have a high frequency of mutations in *TP53*, *RB1* and unstable genomes, whereas NIMBCs have mutations in *FGFR3*, *KDM6A* and stable genomes, and both groups often have *CDKN2A* deletions [262].

Mutations in the Neh2 region of *NFE2L2* have been found in MIBCs (particularly associated with patients that have a history of smoking) and are thought to increase NRF2 activity [158,263]. *KEAP1* is mutated less frequently than *NFE2L2* and the mutations spread throughout the gene. *CUL3* is not mutated in MIBCs but copy number deletions occur [158]. Data from the Cancer Genome Atlas Research Network has shown that *TP53* is commonly mutated in MIBCs [158,262], which again supports the idea that *NFE2L2* mutations may co-occur with mutations in specific driver genes. In the case of UBC, *TP53* mutations are very prevalent in MIBCs, which is the subtype of UBC that have *NFE2L2* mutations. However, there are not enough data to support this association currently.

The stress-inducible scaffold protein p62/SQSTM1 is over-expressed in both UBC cell lines and tissues leading to increased NRF2 activation and protection of the cancer cells against oxidative stress [264]. The phagocytosis-related protein GULP1 is often decreased in UBC cell lines and tissues due to epigenetic alterations. Silencing of GULP1 leads to increased NRF2 activity and the emergence of drug resistance. GULP1 has now been shown to bind directly and stabilize KEAP1 protein locking it bound to NRF2 in the cytoplasm [265]. Also, exposure to several different carcinogens has been linked to the development of bladder cancer and the inactivation and transport of such compounds often involves UGT and GST enzymes, the genes of which are NRF2 targets [261].

NRF2 is upregulated in UBC, specifically in MIBCs which are the late-stage more aggressive invasive form of the disease. This suggests that in UBC, increases in NRF2 activity through somatic mutations occurs in the transition from early-stage to late-stage disease, to support carcinogenesis after it has become established. By contrast, Nrf2-ko mice are more susceptible to carcinogen-induced UBC than are wildtype mice, suggesting that Nrf2 may also have an anti-cancer effect during disease initiation [91] and expression of the Nrf2-target gene *Gstp1* has been implicated in the detoxification of compounds that induced UBC [261].

### 10.7. Upregulation of NRF2 in Colorectal Tumours

Colorectal cancer (CRC) is the 3rd most common cancer worldwide, with more cases being diagnosed each year [214]. 5-Fluorouracil is the current standard treatment for the disease but is associated with high levels of acquired drug resistance [266]. There are several known prognostic factors associated with CRC, including the side of the colon on which the tumour is located, mutational status of driver genes (such as *KRAS* and *BRAF*), epigenetic modifications, genomic instability and DNA mismatch repair status (due to mutations in *MLH1*, *MLH3*, *MSH2*, *MSH3*, *MSH6* and *PSM2*) [267]. Molecular profiling of early- and late-stage metastatic CRC has identified a high frequency of somatic mutations in *APC* (most mutated gene), *KRAS*, *BRAF*, *TP53*, *PIK3CA* and *SMAD4* [159,268].

Mutations in *NFE2L2* rarely occur in CRC and when the mutations are mapped onto the NRF2 protein they are distributed throughout the primary structure, suggesting they probably represent passenger mutations. *KEAP1* mutations are also infrequent and occur throughout the gene. *CUL3* mutations were not present in the dataset [159]. Due to their low frequency in the case of *NFE2L2* mutations and their absence in the case of *CUL3* mutations, their co-occurrence with other mutations in CRC cannot be estimated. Missense mutations in *KEAP1* co-occur with mutations in *BRAF*, *PIK3CA* and *APC* but are mutually exclusive from mutations in *KRAS* and *TP53* [159].

Similar to oesophageal cancer, the high levels of NRF2 activity found in CRC do not correlate with the presence of *KEAP1* and *NFE2L2* mutations, which are relatively rare according to analysis by the Cancer Genome Atlas Research Network of somatic mutations in CRC. Nevertheless, NRF2 is upregulated in CRC tumours, and high levels of nuclear NRF2 protein correlate with poor prognosis [269], suggesting a role in tumour progression. One possible explanation for the upregulation of NRF2 in this instance is DNA methylation: this is an epigenetic process by which an addition of a methyl group to DNA alters gene expression, and if CpG islands located in the gene promoter are methylated, chromatin remodelling may occur that leads to repression of transcription. High levels of methylation of the *KEAP1* gene promoter have been observed in human CRC cell lines and tumour samples, leading to lower levels of KEAP1 mRNA, increased NRF2 protein and over-expression of its target genes [267,270]. Conversely, demethylation of a CpG island in the promoter of *NFE2L2* has been identified in CRC tumour samples. Whereas, the same site has been shown to be methylated in pre-cancerous colorectal polyps suggesting that the demethylation of *NFE2L2* may influence carcinogenesis [271]. As *KRAS* and *BRAF* mutations are prevalent in CRC, NRF2 may frequently be upregulated by transcriptional activation of *NFE2L2*, as reported in mouse models of pancreatic cancer [164].

CRC is thought to be strongly driven by ROS. *APC* mutations that lead to the activation of WNT signalling are one of the earliest initiation events in CRC and are associated with the generation of both the superoxide anion radical (O_2_^●–^) and H_2_O_2_ [272]. Also, in the human colorectal HCT116 cancer cell line, nitric oxide, which is produced during inflammation, has been shown to lead to nuclear accumulation of NRF2, potentially through modification of key reactive cysteine residues in KEAP1 [273]. Consistent with the likely contribution of ROS to the development of CRC, inflammation has been shown to be a significant risk factor, with inflammatory bowel disease patients having a higher chance of developing the disease [272].

High levels of NRF2 activity and overexpression of its target genes are associated with later stage and poorer patient survival in CRC [267,274]. In the clinical setting, this might in part be explained by the fact that high levels of NRF2 have been implicated in the development of resistance against 5-fluorouracil [275]. It is thought that increases in NRF2 activity provide cytoprotection to premalignant adenomatous cells in the early stages of carcinogenesis [275] and that once the cancer is established high levels of NRF2 will support tumour progression.

## 11. Therapeutic Approaches to Treat Tumours in Which NRF2 Is Upregulated

As cancer chemopreventive blocking agents protect normal tissues against initiation of carcinogenesis by inducing NRF2-target genes encoding enzymes that minimise the genotoxic and cytotoxic effects of carcinogens, it is not surprising that the permanent activation of NRF2 in tumour cells confers resistance to therapeutic agents, as well as radiotherapy. Because NRF2 can mediate cancer chemoprevention and also support tumour promotion/progression, it has been referred to as a ‘doubled-edged sword’ [122]. The potentially adverse effects of NRF2 activation in pre-neoplastic lesions and in tumour cells are well recognised and have been referred to by Donna Zhang’s laboratory as the ‘dark side’ of the transcription factor [165,166,167,276]. This is also reflected by the fact that in the clinical setting, upregulation of NRF2 is associated with poor prognosis and decreased overall survival in patients with lung cancer [277], head and neck cancer [254], oesophageal cancer [233], gastric cancer [278], liver cancer [279] and colorectal cancer [267,269].

As cancer cells are potentially more sensitive to oxidative stress than normal cells, because of their high ROS burden caused by oncogene activation, NADPH oxidase activation and mitochondrial dysfunction, therapeutic strategies to augment ROS production or diminish their antioxidant capacity have been considered as a means of producing selective toxicity in tumours [280]. For example, from a large chemical screen, a class of drugs that induced oxidative stress was identified because of their selective toxicity towards *Kras^G12D^*-expressing MEFs, with lanperisone being the most potent and shown to increase ROS production and stimulate non-apoptotic death [281]. In terms of suppressing the GSH-based antioxidant system, inhibition of the GCLC/GCLM heterodimeric enzyme by buthionine sulfoximine, inhibition of GSR1 by 2-acetylamino-3-[4-(2-acetylamino-2-carboxyethylsulfanylthiocarbonylamino)phenylthiocarbamoyl-sulfanyl]propionic acid (2-AAPA), and inhibition of the SLC7A11 antiporter by sulfasalazine or erastin, have been proposed as means of sensitizing tumour cells to chemo- and radiotherapy [282]. In addition, suppressing the TXN-based antioxidant system by targeting TXN using PX-12 and targeting TXNRD using auranofin or motexafin gadolinium have also been proposed to sensitise tumours to radiotherapy.

In tumour cells that utilize heightened levels of ROS to drive proliferation, it has been pointed out that the demands of maintaining high antioxidant levels in order to avoid cell death places pressure on their intracellular pools of glutamate, cysteine and glycine, required to produce GSH, and which in turn results in an increased dependency on an exogenous supply of glutamine, serine and asparagine required to meet such demands [283,284]. Consequently, tumour cells that require high GSH levels for survival are potentially more sensitive than normal cells to glutaminase inhibitors that prevent glutamine from being converted to glutamate, which would in turn be used by SLC7A11 to import cystine into cells [283]. Indeed, these workers have proposed that tumour cells with upregulated NRF2 have increased dependency on exogenous non-essential amino acids, and that depletion of non-essential amino acids in the microenvironment preferentially inhibits proliferation of MEFs lacking Keap1 [284].

The most obvious way of treating tumours harbouring mutations in *NFE2L2* or *KEAP1* is with drugs that inhibit NRF2 activity, and a range of small molecules have been reported to possess this ability. Arguably the best-known example of this strategy is provided by the quassinoid brusatol [285], whilst other examples include retinoic acid [286] or the natural flavonoids luteolin and wogonin [287,288]. These molecules all suffer from issues of specificity and confounding off-target effects, with brusatol inhibiting global protein translation [289]. A number of academic research groups have screened various chemical libraries for small molecules that inhibit NRF2 activity, using a variety of cell-based reporter assays. Using ARE-driven luciferase assays, compounds such as 4-(2-cyclohexylethoxy)aniline [290], ML385 [291], halofuginone [292] and clobetasol propionate [293] have been identified as effective inhibitors of NRF2 activity. It is not clear how many of these will be of value in clinical practice.

Since upregulation of NRF2 increases NADPH levels within cells, reductive bioactivation of drugs might be considered a potentially effective strategy to treat cancer cells harbouring mutations in *NFE2L2*, *KEAP1*, or *CUL3*. Of particular relevance in this regard is the enzyme NQO1, which is often overexpressed in rat liver preneoplastic nodules and human tumours (reviewed in [145,294,295]) where it may be capable of bioreductive activation of quinone-containing xenobiotics. The first compounds recognised to be subject to NQO-mediated activation included 5-(aziridine-1-yl)-2,4-dinitrobenzamide (CB1954, also known as Tretazicar) and mitomycin C, both of which are activated to bifunctional alkylating agents that can cross-link DNA (Figure 6) [296,297,298,299].

Subsequently, it has been found that activation of CB1954 is catalysed by NQO1 in rat cells and NQO2 in human cells [300,301]. Following the demonstration that NQO1 activates mitomycin C, Dave Ross and colleagues discovered that NQO1 is capable of activating a range of antitumour quinones, including streptonigrin, 2,5-diaziridinyl-3-phenyl-1,4-benzoquinone, 2,5-diaziridinyl-3,6-dimethyl-1,4-benzoquinone and [3-hydroxymethyl-5-aziridinyl-1-methyl-2-(1*H*-indole-4,7-dione)-propenol] [302,303]. Some of these compounds are now being investigated in clinical trials [304,305] but more development is necessary for them to be used in the clinic. It should be mentioned that a potentially valuable use of NQO1-bioactivatable agents is their ability to synergise with PARP1 inhibitors and radiotherapy, thereby allowing a reduction in dose of agents administered and minimisation of unwanted toxic side effects [306].

An alternative strategy to exploiting the overexpression of NQO1 to reductively activate antitumour drugs is that of stimulating cytotoxic levels of ROS production, through futile NQO1-catalysed redox-cycling of quinone-containing compounds such as the naturally-occurring 1,2-naphthoquinone β-lapachone [307], and the more potent synthetic compound deoxynyboquinone [308]. Notably, however, despite overexpression of NQO1, cells with upregulated NRF2 may be resistant to β-lapachone due to enhanced detoxification of ROS, though this can be overcome by inhibition of TXNRD or SOD1 [269]. Other deoxynyboquinone-related compounds, DNQ-7 and IB-DNQ, have also been reported to stimulate death in an NQO1-associated manner [295].

The benzoquinone-containing compound 17-N-allylamino-17-demethoxygeldanamycin (17-AAG) and the related 17-DMAG, are examples of drugs that NQO1 activates by catalysing their reduction to a hydroquinone ansamycin, which inhibits heat shock protein 90 (HSP90) and destabilises mutated oncogenic proteins in cancer cells such as protein kinases, steroid receptors and transcription factors [309,310]. Interestingly, it has been found that 17-AAG is significantly more toxic to mouse Hepa1 cells with upregulated Nrf2, from knockout of Keap1, than those in which regulation of the transcription factor by Keap1 is intact, and this increased toxicity is dependent on the quinone moiety within the geldanamycin scaffold because HSP90 inhibitors lacking this structure were not more toxic to cells in which NRF2 is upregulated [311]. It has, therefore, been proposed that reduction of the 17-AAG quinone to the 17-AAGH_2_ hydroquinone, catalysed by overexpressed Nqo1, resulted in more potent inhibition of Hsp90 that resulted in cell death, though other ‘off-target’ explanations are possible. More recently, Liam Baird and colleagues have used their cell-based differential survival screen to identify other small molecules that display selective toxicity towards Hepa1 cells lacking Keap1. Remarkably, from amongst currently-approved drugs, they identified mitomycin C as an agent that displays preferential toxicity towards cells in which Nrf2 is permanently upregulated, and demonstrated that this is likely to be due to overexpression of Nqo1 and enzymes that generate NADPH via the PPP, all of which are encoded by Nrf2-target genes [312]. They also demonstrated that administration of mitomycin C along with 17-AAG led to synergistic toxicity and suggested that as activation of mitomycin C would lead to DNA damage and activation of 17-AAG would lead to proteotoxic stress that the combination represented ‘concurrent synthetic lethality’.

Unfortunately, 17-AAG may be poorly tolerated by patients due to liver and lung toxicity, which results from redox cycling and/or arylating nucleophilic centres within the cell. To avoid such effects, which involves the 19-position on the quinone ring adjacent to one of the carbonyls, a series of 19-substituted benzoquinone ansamycins (19-BQAs) have been created [313]. These were found not to react with GSH, were subject to less redox cycling than 17-DMAG and exhibited less hepatotoxicity than DMAG [314]. It remains to be seen whether 19-BQAs are more toxic to cancer cells in which NRF2 is upregulated.

## 12. Concluding Comments

Herein, we have described the background to the discovery that NRF2 is responsible for intrinsic resistance to many chemical carcinogens and that through its ability to mediate cellular adaption to oxidative and electrophilic stress it orchestrates induction of cytoprotective detoxification genes by cancer chemopreventive agents, thus inhibiting initiation of carcinogenesis in stomach, bladder, skin, GI tract, breast, lung and liver. We also recount events that led to the recognition that NRF2 is frequently upregulated in tumour cells, and describe a wide range of mechanisms that allow it to escape repression by KEAP1, with somatic mutations in *NFE2L2*, *KEAP1* and *CUL3* being the best characterized examples. From screening work done through TGCA, it has become clear that NRF2 upregulation is a feature of many common cancers, including those of lung, oesophagus, liver and head and neck. Whilst it might have been anticipated that tissues in rodents in which chemopreventive agents protect against carcinogenesis would be the same as those in which NRF2 is permanently upregulated in human tumours, because NRF2 function is necessary for adaptation in such tissues, this was found to be only partially true. There are at least three reasons for this disparity: firstly, the NRF2-mediated mechanisms responsible for chemoprevention (increased detoxification) differ from those NRF2 confers during tumour progression (increased ROS scavenging, generation of NADPH, synthesis of serine and synthesis of ribonucleotides); secondly, as NRF2 can be upregulated by mechanisms other than somatic mutations in *NFE2L2*, *KEAP1* and *CUL3*, the full extent to which NRF2 is constitutively activated in cancer is not known; thirdly, the demands placed on NRF2 to support growth and survival of cells harbouring mutations in genes that drive tumourigenesis are highly variable across different cancer types.

Evidence from studies of the rat Solt-Farber hepato-carcinogenesis model and clinical NSCLC samples suggest that upregulation of NRF2 occurs early during tumourigenesis, and that its constitutive activation probably supports the actions of oncogenic KRAS and hyperactivated PKB/Akt. However, whilst these studies might be interpreted to indicate NRF2 upregulation is less important in late stages of cancer than in the early promotion/progression stages, this does not seem to be universally true. For example, mutations in *KEAP1* and *NFE2L2* have been reported in late-stage bladder cancer and NRF2 has been reported to be upregulated in a model of post-therapy recurrence of breast cancer. Moreover, whilst it is well recognised that NRF2 upregulation benefits tumours because it provides protection against ROS-stimulated cell death and supports cell proliferation during all stages of cancer evolution, its putative ability to modulate the innate immune system may be important in late stage cancer. Clearly, more work is required to establish why NRF2 upregulation occurs in later progression and metastatic stages of disease.

As upregulation of NRF2 frequently confers upon tumours resistance to chemo- and radio-therapy, strategies are urgently required to overcome such resistance. To this end, a variety of ROS-based and quinone-based pharmacological approaches that capitalise on overexpression of NRF2-target genes are described, but more developmental work is required to generate drugs that are clinically useful. The functional and regulatory inter-relationships between NRF2 and other antioxidant transcription factors such as AP-1, FOXO, PGC-1α, NF-κB and TP53 during the various stages of tumourigenesis is not understood, and this issue needs to be addressed for us to appreciate fully the importance of redox signaling during the different stages of tumourigenesis; this is likely to be important in tumours with mutant *TP53*. Understanding the contributions made by NRF2 to tumourigenesis has enabled new diagnostic and prognostic tools to be developed and provides a strong rationale for effective therapeutic strategies to overcome drug resistance.

One of the major obstacles in studying the course of events that occurs during the evolution of cancer is the large amount of cellular heterogenicity observed in tumours. Previously, transcriptomic analysis has been limited to examining populations of cells, but the development of single cell sequencing technology now allows the analysis of individual cells and has been used to address the question of heterogenicity in several cancer types [315,316]. Studying the impact of mutations in *NFE2L2*, *KEAP1* and *CUL3* will be made possible through the use of precise techniques such as Base Editors [317] and Saturation Genome Editing [318], which can be used to generate specific point mutations. These techniques, in combination with high-throughput screening technologies, will allow the field to address important questions surrounding the impact of mutations in *NFE2L2*, *KEAP1* and *CUL3*. Currently, studies using these techniques are limited due to the high associated cost, but hopefully as the technology develops and the cost reduces, such technology will become more accessible to researchers in the field [316]

## Figures and Tables

**Figure 1 cancers-12-03609-f001:**
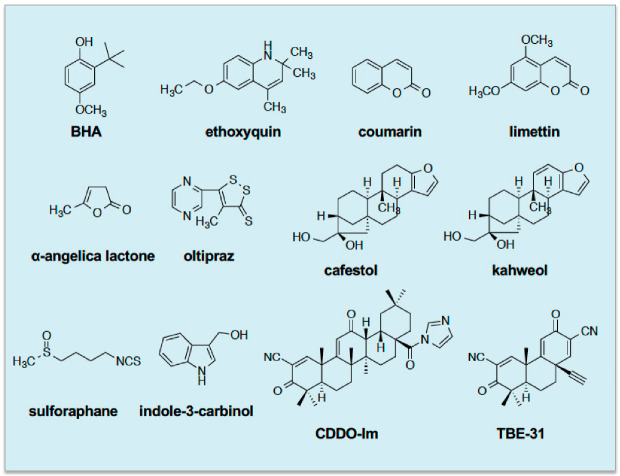
Structures of xenobiotics that require Nrf2 to induce mouse *Nqo1* and *Gst* genes. These include butylated hydroxyanisole (BHA), ethoxyquin, coumarin, limettin, α-angelica lactone, oltipraz, cafestol, kahweol, sulforaphane, indole-3-carbinol, 1-[2-cyano-3,12-dioxooleana-1,9(11)-dien-28-oyl]imidazole (CDDO-Im) and TBE-31.

**Figure 2 cancers-12-03609-f002:**
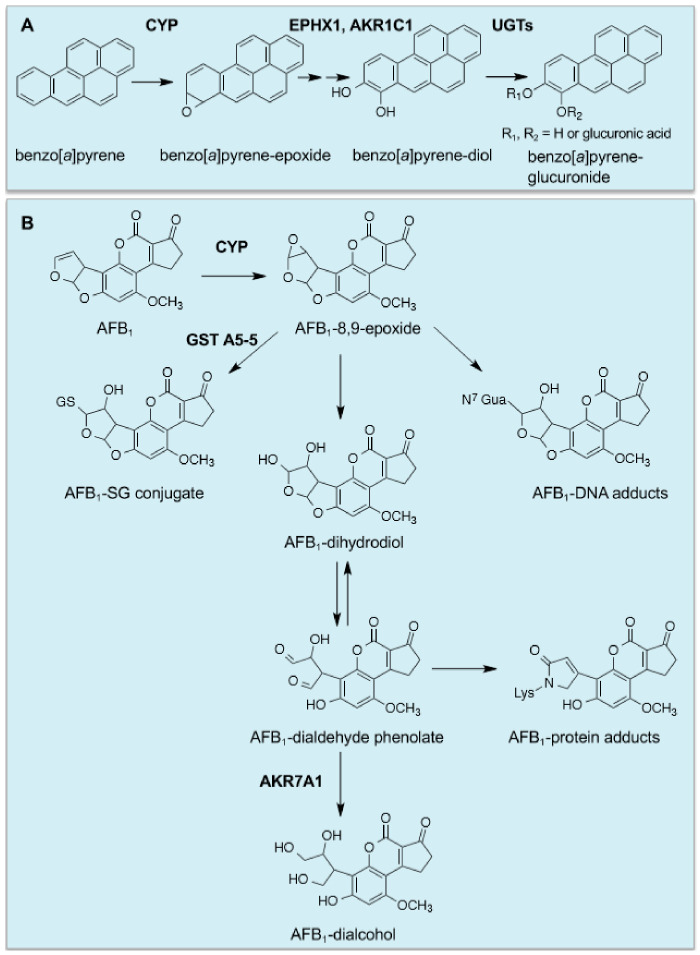
Chemopreventive agents stimulate the metabolic inactivation of chemical carcinogens. The cytochrome P450 (CYP)-catalysed activation of benzo[*a*]pyrene to the epoxidated ultimate carcinogen and its metabolism by EPHX1, AKR1C1 and UGTs is shown in panel (**A**). The CYP-catalysed activation of aflatoxin B_1_ to the epoxidated ultimate carcinogen and its metabolism by GST A5-5 and AKR7A1 is shown in panel (**B**).

**Figure 3 cancers-12-03609-f003:**
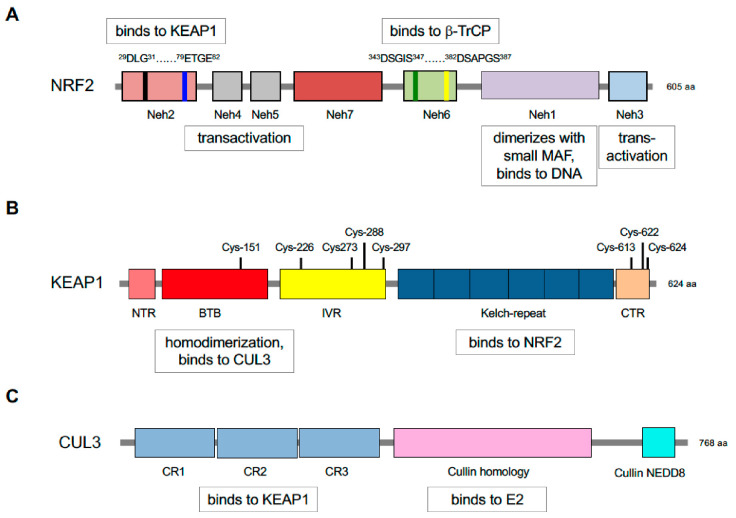
Domain structure of NRF2, KEAP1 and CUL3. The cartoon depicts: (**A**) the positions of the Neh1-Neh7 domains in human NRF2, along with amino acids associated with the DLG^31^ and ETGE^82^ motifs and their respective functions; (**B**) the positions of the NTR, BTB, IVR, Kelch-repeat and CTR domains in KEAP1; (**C**) the positions of the cullin repeat 1 (CR1), CR2, CR3 and cullin homology (CH) domains in CUL3.

**Figure 4 cancers-12-03609-f004:**
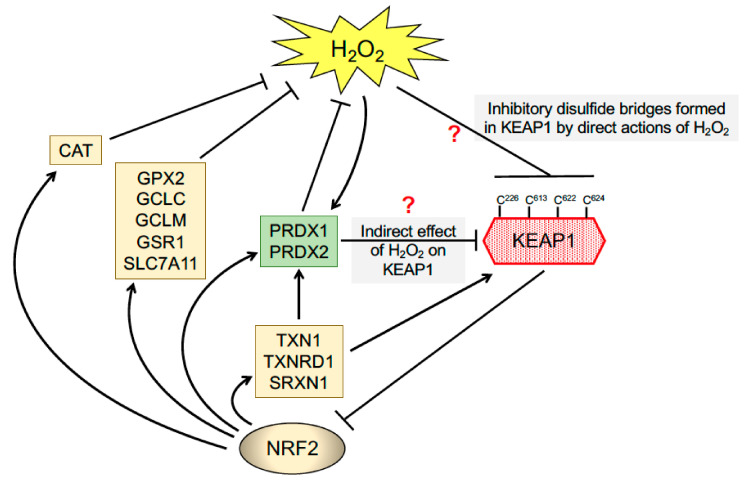
Stimulation of the NRF2-KEAP1 axis by H_2_O_2._ The cartoon depicts potential ways in which KEAP1 is inactivated by H_2_O_2_ and how induction of NRF2-target genes suppresses intracellular H_2_O_2_ levels. In particular, it draws attention to the possibility that PRDX1 and PRDX2 might be the primary sensor of H_2_O_2_ and when oxidised, relay this information to KEAP1 by stimulating disulfide bridge formation between Cys226, Cys613, Cys622 and Cys624 in the ubiquitin ligase substrate adaptor, allowing NRF2 to evade ubiquitylation by CRL^KEAP1^ and induce antioxidant genes (shown in boxes), which in turn serve to supress intracellular H_2_O_2_ levels. However, through induction of the TXN-based antioxidant system, activated NRF2 would also stimulate reduction of disulfide bonds in KEAP1, and so restore repression of NRF2 by CRL^KEAP1^, thereby providing a negative feedback loop.

**Figure 5 cancers-12-03609-f005:**
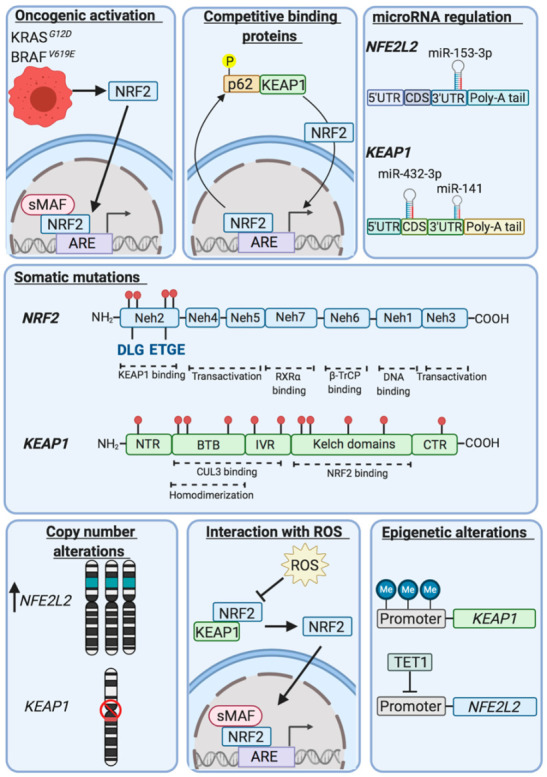
Mechanisms of upregulation of NRF2 in tumours. The cartoon shows the various ways in which NRF2 activity can be upregulated in cancer. The top left-hand panel shows how NRF2 activity can be increased by oncogenic proteins: the middle top panel shows how p62/SQSTM1 binds to KEAP1, thereby preventing KEAP1 from repressing NRF2; the top right-hand panel indicates how microRNAs can increase NRF2 activity through interacting with either *NFE2L2* or *KEAP1* transcripts. The center panel illustrates regions of *KEAP1* and *NFE2L2* that are frequently somatically mutated in cancer (with mutations represented by red circles). The bottom left-hand panel depicts how NRF2 abundance can be enhanced by increases in *NFE2L2* copy number or decreases in *KEAP1* copy number; the bottom center panel depicts how interaction with ROS oxidize cysteine residues in KEAP1 preventing it from targeting NRF2 for proteasomal degradation; the bottom right-hand panel highlights how epigenetic alterations in *KEAP1* and *NFE2L2* impact NRF2 abundance.

**Figure 6 cancers-12-03609-f006:**
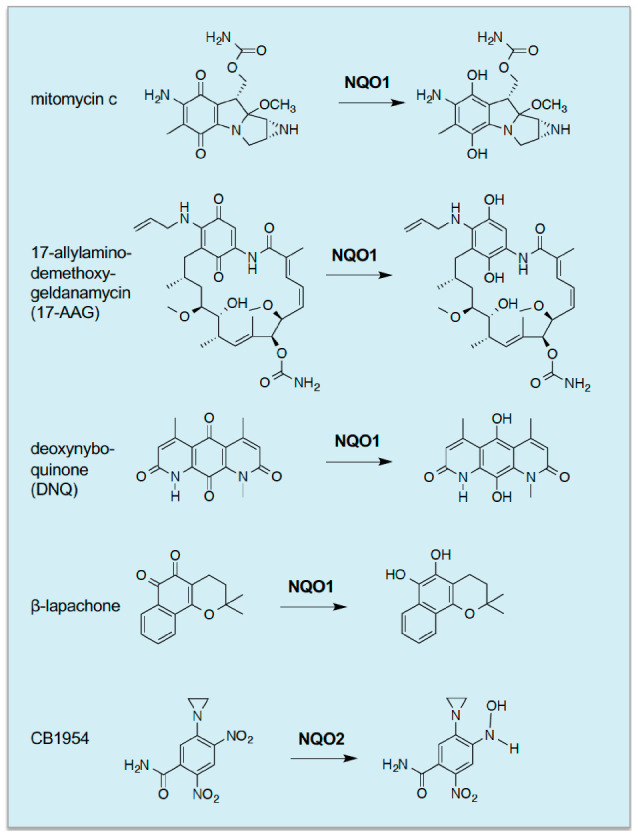
Chemicals activated by NQO1 and NQO2. The structures of mitomycin C, 17-allylaminodemethoxygeldanamycin, deoxynyboquinone, β-lapachone and CB1954 (Tretazicar), which are bio-reductively activated by NQO1 and NQO2, are shown.

**Table 1 cancers-12-03609-t001:** NRF2-target genes.

Function	Gene
Phase I drug oxidation, reduction and hydrolysis	*AKR1B10*, *AKR1C1*, *AKR1C2*, *AKR1C3, ALDH1A1*, *ALDH3A1*, *ALDH7A1*, *CBR1*, *CES1G*, *CES1H*, *EPHX1*, *NQO1*, *PTGR1*
Phase II drug conjugation *	*GSTA1*, *GSTA3*, *GSTM1*, *GSTP1*, *MGST1*, *SULT1A1*, *UGT1A1*, *UGT1A6*, *UGT2B7*
Phase III drug efflux	*ABCB6*, *ABCC1*, *ABCC2*, *ABCC3*
Scavenging of H_2_O_2_	*CAT*, *GPX2*, *GPX4*, *PRDX1*, *PRDX6*
Iron/heme metabolism	*BLVRB*, *FECH*, *FPN1* (also called *SLC40A1*), *FTH1*, *FTL1*, *HMOX1*
Contribution to maintenance of the GSH-based antioxidant system	*GCLC*, *GCLM*, *GGT1*, *GSR1*, *SLC7A11*
Contribution to maintenance of the TXN1-based antioxidant system	*TXN1*, *TXNRD1*, *SRXN1*
NADPH generation to provide reducing equivalents for GSH and TXN1 systems	*G6PD*, *PGD*, *IDH1*, *ME1*
Non-oxidative arm of the pentose phosphate pathway **	*TALDO1*, *TKT*
Import of fatty acids	*CD36*
Scavenger receptor	*MARCO*
Proteasomal subunits	*PSMA1*, *PSMA4*, *PSMA7*, *PSMB1*, *PSMB2*, *PSMB3*, *PSMB4*, *PSMB5*, *PSMB6*, *PSMC1*, *PSMC3*, *PSMD1*, *PSMD5*, *PSMD7*, *PSMD11*, *PSMD12*, *PSMD13*, *PSME1* (encoding PA28αβ)
Autophagy-related proteins	*SQSTM1* (encoding p62), *CALCOCO2*, *ULK1*, *ATG2*, *ATG5*, *ATG7*, *GABARAPL1*, *LAMP2*
Transcription factors and associated factors	*AHR*, *ATF3*, *ATF4*, *CEBPB* (encoding C/EBPβ), *MAFG*, *PPARG* (encoding PPARγ, or NR1C3)*, PPARGC1B, RXRA* (encoding RXRα, or NR2B1)
Ubiquitin ligase substrate adaptor	*KEAP1*
Positive regulation of response to oxidative stress	*NFE2L2*
Activation of Ca^+2^ channel signalling	*TRPA1*
Activation of Notch signalling	*NOTCH1*
Attenuation of hedgehog signalling ***	*PTCH1*

* cytosolic glutathione S-transferase (GSTs) induced in rat and mouse, and not the human; ** requires growth factor signaling; *** both positive and negative control have been reported.

**Table 2 cancers-12-03609-t002:** Frequency of mutations in *KEAP1*, *NFE2L2* and *CUL3* across different cancer types.

Cancer Type	Percentage of Somatic Mutation			Reference	Samples Analysed
	*KEAP1*	*NFE2L2*	*CUL3*		
Lung	14.2%	7.3%	3.4%	[153]	Whole-exome sequencing of 660 LUAD and 484 LUSC tumour/normal pairs
Oesophageal	3.4%	4.5%	1.1%	[154]	Whole-genome or whole-exon sequencing of 88 ESCC tumour/normal pairs
Liver	5.1%	4.0%	0%	[155]	Hepatocellular carcinomas 369 and 3 fibrolamellar carcinoma
Head and neck	4.3%	6.1%	3.6%	[156]	Whole exome sequencing and/or whole genome sequencing of 279 head and neck squamous cell carcinoma tumour/normal pairs
Gastric	3.1%	0.3%	2.0%	[157]	Whole exome sequencing of 295 primary gastric adenocarcinomas tumours with matched normal
Bladder	3.1%	8.7%	0%	[158]	Whole exome sequencing of 131 high grade muscle invasive urothelial bladder carcinomas
Colorectal	1.4%	0.9%	0%	[159]	Whole exome sequencing in 224 of the 276 colorectal carcinoma tumour/normal pairs

This table was created using Cbioportal [165,166].

**Table 3 cancers-12-03609-t003:** Mutations in genes implicated in regulating NRF2 activity in NSCLC *.

Pan-Lung Cancer		
	Alteration	
Gene Name	Number of Samples with the Mutation	Number of Samples without the Mutation
*NFE2L2*	105 (9%)	1039 (91%)
*KEAP1*	170 (15%)	974 (85%)
*CUL3*	51 (5%)	1093 (96%)
*KRAS*	259 (23%)	885 (77%)
*TP53*	776 (68%)	368 (32%)
*PIK3CA*	276(24%)	868 (76%)
*PTEN*	102 (9%)	1042 (91%)
*STK11*	118 (10%)	1026 (90%)

* Data from [153]. This table was created using Cbioportal [165,166].

**Table 4 cancers-12-03609-t004:** *KEAP1* mutations occur alongside mutations in other genes in NSCLC *.

*KEAP1* Mutant Samples			
Gene	Alteration		Co-Occurrence or Mutual Exclusivity with *KEAP1* Mutation
	Number of *KEAP1* Mutant Samples with the Alteration	Number of *KEAP1* Wildtype Samples with the Alteration	
*NFE2L2*	4(2.4%)	80 (8.2%)	Mutual exclusivity
*CUL3*	6 (3.5%)	33(3.4%)	Co-occurrence
*KRAS*	41 (24.1%)	181 (18.6%)	Co-occurrence
*TP53*	103 (60.6%)	672 (69%)	Mutual exclusivity
*PIK3CA*	12 (7.1%)	82(8.4%)	Mutual exclusivity
*PTEN*	6 (3.5%)	61(6.3%)	Mutual exclusivity
*STK11*	39 (22.9%)	72 (7.4%)	Co-occurrence

* Data from [153]. This table was created using Cbioportal [165,166].

**Table 5 cancers-12-03609-t005:** *NFE2L2* mutations occur alongside mutations in other genes in NSCLC *.

*NFE2L2* Mutant Samples			
Gene	Alteration		Co-Occurrence or Mutual Exclusivity with *NFE2L2* Mutation
	Number of *NFE2L2* Mutant Samples with the Alteration	Number of *NFE2L2* Wildtype Samples with the Alteration	
*KEAP1*	6(6.7%)	155(15%)	Mutual exclusivity
*CUL3*	5 (4.8%)	34(3.3%)	Co-occurrence
*KRAS*	5 (4.8%)	217 (20.9%)	Mutual exclusivity
*TP53*	88 (83.8%)	687 (66.1%)	Co-occurrence
*PIK3CA*	12 (11.4%)	82 (7.9%)	Co-occurrence
*PTEN*	5 (4.8%)	62 (6%)	Mutual exclusivity
*STK11*	3 (2.9%)	108 (10.4%)	Mutual exclusivity

* Data from [153]. This table was created using Cbioportal [165,166].

**Table 6 cancers-12-03609-t006:** *CUL3* mutations occur alongside mutations in other genes in NSCLC *.

*CUL3* Mutant Samples			
Gene	Alteration		Co-Occurrence or Mutual Exclusivity with *CUL3* Mutation
	Number of *CUL3* Mutant Samples with the Alteration	Number of *CUL3* Wildtype Samples with the Alteration	
*NFE2L2*	5 (9.8%)	79 (7.2%)	Co-occurrence
*KEAP1*	6 (11.8%)	156 (14.3%)	Mutual exclusivity
*KRAS*	4 (7.8%)	218 (20%)	Mutual exclusivity
*TP53*	42 (82.4%)	733 (67.1%)	Co-occurrence
*PIK3CA*	6 (11.8%)	88 (8.1%)	Co-occurrence
*PTEN*	4 (7.8%)	63 (5.8%)	Co-occurrence
*STK11*	3 (5.9%)	108 (9.9%)	Mutual exclusivity

* Data from [153]. This table was created using Cbioportal [165,166].

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
