# Peer review of "NRF2 and the Ambiguous Consequences of Its Activation during Initiation and the Subsequent Stages of Tumourigenesis"

_cancers, 2020, doi:10.3390/cancers12123609_

Round 1
Reviewer 1 Report
The authors of this review entitled Nrf2 and the ambiguous consequences of its activation during the different stages of tumourigenesis discuss and summarize current knowledge concerning the role of transcription factor NRF2 in tumorigenesis and indicate its therapeutic value.
Comments:
The paper is quite long and in my opinion should be rewritten. The main topic of the manuscript concerns the NRF2 role in cancer chemoprevention and treatment. According to that the authors and editors of the journal should consider the removal of paragraphs from 7 to 9. Then the paper will be more consistent and clear for the readers.
The paper includes a lot of editorial errors:
- line 5, John D. Hayes 1 and *
- the “Simple Summary”, “Keywords”, “Funding”, “Author Contributions” and “Conflicts of Interest” sections are missed
- resolution of the figures is quite poor
Author Response
The aim of the article was to form conclusions about the impact of NRF2 on cancer by drawing on chemoprevention studies and clinical data from the TCGA. It was never intended to be solely about chemoprevention. We have therefore extensively revised the text to make the link between cancer chemoprevention in experimental animals with studies by TCGA of somatic mutations in clinical cancer. Furthermore, we have removed the section entitled “Perceived risk of antioxidant therapies”, which was on pages 35 and 36 of the original manuscript, as it is of marginal relevance to the main theme. Importantly, we have discussed in the “Concluding comments” section, on pages 38 and 39, how the results from animal cancer chemoprevention relate to clinical findings, and what inferences can be drawn.
A deficiency in our original article was the lack of information about the mechanisms by which NRF2 activation supports tumourigenesis. We have therefore added a new section entitled “Mechanisms by which upregulation of NRF2 supports post-initiation stages of cancer” on pages 21 to 23, which addresses this point. We have also significantly extended Table 1 to include all NRF2-regulated genes that are thought to support tumourigenesis.
We thank the reviewer for spotting the missing "and", and it is now added.
We thank the reviewer for mentioning the simple summary, which we have supplied on page 57. Also, we thank them for mentioning key words, source of funding, author contributions, and conflict of interest, which are now supplied on page 58.
We have supplied new versions of Figures 3, 4 and 5 as we realise the resolution of the originals may have been suboptimal. Figures 1, 2 and 6 are chemical structures, and they seem good enough to us.
Reviewer 2 Report
In the manuscript titled "Nrf2 and the ambiguous consequences of its activation during the different stages of tumorigenesis" prepared by Robertson et al., the author contributed a systemic overview of the role of Nrf2 throughout tumorigenesis. Overall I think the article is well prepared and provides timely updates to the field. I have only a few minor suggestions.
- The author reviewed the roles of Nrf2 in several types of solid tumors, including lung, oesophageal, liver, head&neck, gastric, bladder, and colorectal cancers. I recall that there are emerging studies about Nrf2 in brain tumors in this couple of years. The author may consider updating this type of solid tumor so that the article could reach a wider audience. For example, the Nrf2-Keap1 pathway promotes cellular proliferation in GBM cells by suppressing ferroptosis (PMID: 28805788). Nrf2-derived glutathione metabolism is critical for glioma cells with metabolic stress (PMID: 30759236). Nrf2 inhibition enhances the therapeutic efficacy of chemotherapy to glioma cells through Ras/Raf/MEK pathway (PMID: 32378973).
- The author may go through the article for consistency in abbreviations. For example, "Nrf2" is used in the title, whereas "NRF2" is used in most parts of the article.
- In the title page, I think "and *" should be superscripted.
- Same in the title page, the "Simple Summary" is apparently from the template, but not relevant to the present work.
- Same in the title page, Keywords are still the content from template.
- In line 52, the "see below" is a bit confusing, whether this is referring to the content or the figure?
- In line 168, some of the references were labeled "57 1980". I am not sure if this meets the publication requirements.
- In lines 1036-1037, some of the funding information remains "XXX".
- In line 1047, the reference is labeled "290 2013".
Author Response
We thank the reviewer for their helpful suggestions. We have answered the points raised individually.
Point 1. The idea that we include glioma in the review is very interesting, as is the mention of ferroptosis. We have therefore added a paragraph about this on page 19.
Point 2. We tried to use "NRF2" to describe the transcription factor in a general sense within the text, and "Nrf2" when talking about the mouse. There were mistakes, and we thank the reviewer for spotting this inconsistency.
Point 3. We are puzzled by the superscripted comment, and don't know how to interpret it. Sorry, please clarify if appropriate.
Point 4. The simple summery has been moved to the back of the m/s, and is on page 57.
Point 5. A fuller list of keywords is now on page 58.
Point 6. The words "see below" on line 52 have been removed.
Point 7. The reference on line 168 has been corrected. Thanks
Point 8. Funding information has been supplied on page 58.
Point 9. On line 1047 the reference has been corrected. Thanks
Reviewer 3 Report
NRF2 is a versatile protein in cancer, which either facilitates the emergence and progression of cancer, or on the contrary restrict its development. This context- and/or tissue-dependent effect remains for the most part not understood, making the literature on NRF2 often confusing, and rather complex. The review by Robertson and colleagues perfectly grasps this complexity. It is very well written, very informative and it clearly addresses the limitations in our understanding of NRF2 function in cancer and its influence on the response to chemotherapies. It would be interesting to get a little bit more information and probably guesses from the authors on how development of new technologies could help in better understanding the function of NRF2 in cancer.
Minor:
-several sections of the manuscript are not written: keywords, conflict of interest and funding.
- few typos here and there, for ex. in the list of authors.
- Also fix : « an » on lane 397.
- Page 7 : the abbreviation of 2-AAPA should be in the text.
Author Response
We thank the reviewer for evaluating our manuscript and were particularly heartened that it was scored highly. Much appreciated. We have responded to their comments as follows.
Point 1. Details of keywords, funding and conflict of interest are on page 58. Thanks for spotting this.
Point 2. We have added the word "and" to the author list, but don't believe any of the author's names have been mis-spelt.
Point 3. We have corrected the "an" on line 397.
Point 4. We have taken the full name for the abbreviation 2-APAA from the footnote and placed it within the text.
Round 2
Reviewer 1 Report
However the paper is much longer after correction than before it is now much more interesting. Authors include a lot of additional information which made the manuscript interesting for the readers. Although the text is quite long it is the review paper and as long as it contains essential data it could be valuable.